# FASTER FEDERATED OPTIMIZATION UNDER SECOND-ORDER SIMILARITY

**Ahmed Khaled**
Princeton University

**Chi Jin**
Princeton University

## ABSTRACT

Federated learning (FL) is a subfield of machine learning where multiple clients try to collaboratively learn a model over a network under communication constraints. We consider finite-sum federated optimization under a second-order function similarity condition and strong convexity, and propose two new algorithms: SVRP and Catalyzed SVRP. This second-order similarity condition has grown popular recently, and is satisfied in many applications including distributed statistical learning and differentially private empirical risk minimization. The first algorithm, SVRP, combines approximate stochastic proximal point evaluations, client sampling, and variance reduction. We show that SVRP is communication efficient and achieves superior performance to many existing algorithms when function similarity is high enough. Our second algorithm, Catalyzed SVRP, is a Catalyst-accelerated variant of SVRP that achieves even better performance and uniformly improves upon existing algorithms for federated optimization under second-order similarity and strong convexity. In the course of analyzing these algorithms, we provide a new analysis of the Stochastic Proximal Point Method (SPPM) that might be of independent interest. Our analysis of SPPM is simple, allows for approximate proximal point evaluations, does not require any smoothness assumptions, and shows a clear benefit in communication complexity over ordinary distributed stochastic gradient descent.

## 1 INTRODUCTION

Federated Learning (FL) is a subfield of machine learning where many clients (e.g. mobile phones or hospitals) collaboratively try to solve a learning task over a network without sharing their data. Federated Learning finds applications in many areas including healthcare, Internet of Things (IoT) devices, manufacturing, and natural language processing tasks (Kairouz et al., 2019; Nguyen et al., 2021; Liu et al., 2021). One of the central problems of FL is federated or distributed optimization. Federated optimization has been the subject of intensive ongoing research effort over the past few years (Wang et al., 2021). The standard formulation of federated optimization is to solve a minimization problem:

$$\min_{x \in \mathbb{R}^d} \left[ f(x) = \frac{1}{M} \sum_{m=1}^{M} f_m(x) \right], \tag{1}$$

where each function $f_m$ represents the empirical risk of model $x$ calculated using the data on the $m$-th client, out of a total of $M$ clients. Each client is connected to a central server tasked with coordinating the learning process. We shall assume that the loss on each client is $\mu$-strongly convex.

Because the model dimensionality $d$ is often large in practice, the most popular methods for solving Problem (1) are first-order methods that only access gradients and do not require higher-order derivative information. Such methods include distributed (stochastic) gradient descent, FedAvg (also known as Local SGD) (Konečný et al., 2016), FedProx (also known as the Stochastic Proximal Point Method) (Li et al., 2020b), SCAFFOLD (Karimireddy et al., 2020b), and others. These algorithms typically follow the *intermittent-communication framework* (Woodworth et al., 2021): the optimization process is divided into several communication rounds. In each of these rounds, the server sends a model to the clients, they do some local work, and then send back updated models. The server aggregates these models and starts another round.

Problem (1) is an example of the well-studied finite-sum minimization problem, for which we have tightly matching lower and upper bounds (Woodworth & Srebro, 2016). The chief quality that

differentiates federated optimization from the finite-sum minimization problem is that we mainly care about *communication complexity* rather than the number of gradient accesses. That is, we care about the number of times that each node communicates with the central server rather than the number of local gradients accessed on each machine. This is because the cost of communication is often much higher than the cost of local computation, as Kairouz et al. (2019) state: *"It is now well-understood that communication can be a primary bottleneck for federated learning."*

One of the main sources of this bottleneck is that when all clients participate in the learning process, the cost of communication can be very high (Shahid et al., 2021). This can be alleviated in part by using *client sampling* (also known as *partial participation*): by sampling only a small number of clients for each round of communication, we can reduce the communication burden while retaining or even accelerating the training process (Chen et al., 2022).

Our main focus in this paper is to develop methods for solving Problem (1) using client sampling and under the following assumption:

**Assumption 1.** *(Second-order similarity). We assume that for all $x, y \in \mathbb{R}^d$ we have*

$$\frac{1}{M} \sum_{m=1}^{M} \|\nabla f_m(x) - \nabla f(x) - [\nabla f_m(y) - \nabla f(y)]\|^2 \leq \delta^2 \|x - y\|^2.$$

Assumption 1 is a slight generalization of the $\delta$-relatedness assumption used by Arjevani & Shamir (2015) in the context of quadratic optimization and by Sun et al. (2022) for strongly-convex optimization. It is also known as *function similarity* (Kovalev et al., 2022). Assumption 1 holds (with relatively small $\delta$) in many practical settings, including statistical learning for quadratics (Shamir et al., 2014), generalized linear models (Hendrikx et al., 2020), and semi-supervised learning (Chayti & Karimireddy, 2022). We provide more details on the applications of second-order similarity in Appendix B. Under the *full participation* communication model where all clients participate each iteration, several methods can solve Problem (1) under Assumption 1, including ones that tightly match existing lower bounds (Kovalev et al., 2022). In contrast, for the setting we consider (partial participation or client sampling), no lower bounds are known. The main question of our work is:

*Can we design faster methods for federated optimization (Problem (1)) under second-order similarity (Assumption 1) using client sampling?*

## 1.1 CONTRIBUTIONS

We answer the above question in the affirmative and show the utility of using client sampling in optimization under second-order similarity for strongly convex objectives. Our main contributions are as follows:

- **A new algorithm for federated optimization (SVRP, Algorithm 2).** We develop a new algorithm, SVRP (Stochastic Variance-Reduced Proximal Point), that utilizes client sampling to improve upon the existing algorithms for solving Problem 1 under second-order similarity. SVRP has a better dependence on the number of clients $M$ in its communication complexity than all existing algorithms (see Table 1), and achieves superior performance when the dissimilarity constant $\delta$ is small enough. SVRP trades off a higher computational complexity for less communication.
- **Catalyst-accelerated SVRP.** By using Catalyst (Lin et al., 2015), we accelerate SVRP and obtain a new algorithm (Catalyzed SVRP) that improves the dependence on the effective conditioning from $\frac{\delta^2}{\mu^2}$ to $\sqrt{\frac{\delta}{\mu}}$. Catalyzed SVRP also has a better convergence rate (in number of communication steps, ignoring constants and logarithmic factors) than all existing accelerated algorithms for this problem under Assumption 1, reducing the dependence on the number of clients multiplied by the effective conditioning $\sqrt{\frac{\delta}{\mu}}$ from $\sqrt{\frac{\delta}{\mu}}M$ to $\sqrt{\frac{\delta}{\mu}}M^{3/4}$ (see Table 1).

While both SVRP and Catalyzed SVRP achieve a communication complexity that is better than algorithms designed for the standard finite-sum setting (like SVRG or SAGA), the computational complexity is a lot worse. This is because we tradeoff local computation complexity for a reduced communication complexity. Additionally, both SVRP and Catalyzed SVRP are based upon a novel combination of variance-reduction techniques and the stochastic proximal point method (SPPM). SPPM is our starting point, and we provide a new analysis for it that might be of independent interest. Our analysis of SPPM is simple, allows for approximate evaluations of the proximal operator, and

| Method | Communication Complexity | Non-quadratic? | Reference |
|:---:|:---:|:---:|:---:|
| DANE | $\tilde{\mathcal{O}}\left(\frac{\delta^2}{\mu^2}M\right)$ | ✗ | (Shamir et al., 2014) |
| DiSCO | $\tilde{\mathcal{O}}\left(\sqrt{\frac{\delta}{\mu}}M\right)$ | ✗ | (Zhang & Lin, 2015) |
| SCAFFOLD | $\tilde{\mathcal{O}}\left(\frac{\delta+L}{\mu}M\right)$ | ✗ | (Karimireddy et al., 2020b) |
| SONATA | $\tilde{\mathcal{O}}\left(\frac{\delta}{\mu}M\right)$ | ✓ | (Sun et al., 2022) |
| Accelerated SONATA | $\tilde{\mathcal{O}}\left(\sqrt{\frac{\delta}{\mu}}M\right)$ | ✓ | (Tian et al., 2022) |
| Accelerated Extragradient | $\tilde{\mathcal{O}}\left(\sqrt{\frac{\delta}{\mu}}M\right)$ | ✓ | (Kovalev et al., 2022) |
| Lower Bound (no sampling) | $\tilde{\mathcal{O}}\left(\sqrt{\frac{\delta}{\mu}}M\right)$ | - | (Arjevani & Shamir, 2015) |
| **SVRP** | $\tilde{\mathcal{O}}\left(\frac{\delta^2}{\mu^2}+M\right)$ | ✓ | **Theorem 2 (NEW)** |
| **Catalyzed SVRP** | $\tilde{\mathcal{O}}\left(\sqrt{\frac{\delta}{\mu}}M^{3/4}+M\right)$ | ✓ | **Theorem 3 (NEW)** |

Table 1: Communication complexity of different methods for solving Problem 1 under $\mu$-strong convexity, $L$-smoothness, and Assumption 1. $\tilde{\mathcal{O}}(\cdot)$ ignore polylogarithmic factors and constants. We use the client sampling model, which counts exchanging a vector between the server and one of the clients as a single communication step.

extends to include variance reduction. In Appendix H we also consider the more general constrained optimization problem and provide similar convergence rates in that setting.

## 2 RELATED WORK

**Distributed optimization under Assumption 1.** There is a long line of work analyzing distributed optimization under Assumption 1 and strong convexity: Shamir et al. (2014) first gave DANE and analyzed it for quadratics, and showed the benefits of using second-order similarity in the setting of statistical learning for quadratic objectives. Zhang & Lin (2015) developed the DiSCO algorithm that improved upon DANE for quadratics, and also analyzed it for self-concordant objectives. Arjevani & Shamir (2015) gave a lower bound that matched the rate given by DANE, though without allowing for client sampling. The theory of DANE was later improved in (Yuan & Li, 2019), allowing for local convergence for non-quadratic objectives. Another algorithm SCAFFOLD (Karimireddy et al., 2020b) can be seen as a variant of DANE and is also analyzed for quadratics. In the context of decentralized optimization, Sun et al. (2022) gave SONATA and showed a similar rate to DANE but for general strongly convex objectives, then Tian et al. (2022) improved the convergence rate of SONATA by acceleration. Finally, Kovalev et al. (2022) improved the convergence rate of accelerated SONATA even further by removing extra logarithmic factors. We give an overview of all related results in Table 1 and provide more thorough comparisons in the theory and algorithms section.

A parallel line of work considers Assumption 1 as a special case of *relative strong convexity and smoothness* and utilizes methods based on mirror descent. Hendrikx et al. (2020) take this view and consider an accelerated variant of mirror descent in the distributed setting, while Dragomir et al. (2021) also consider sampling and variance-reduction. Because this setting is much more general, it is more challenging to prove tight convergence rates, and in the worst-case the convergence rates are no better than the minimax rate under only smoothness and strong convexity.

Another line of work considers federated optimization for Problem 1 under Assumption 1 but *without* convexity, as well as optimization under the scenario where the number of clients is very large or infinite. Examples of this include MIME (Karimireddy et al., 2020a), FedShuffleMVR (Horváth et al., 2022), as well as AuxMom and AuxMVR (Chayti & Karimireddy, 2022). Our focus in this work is on the convex setting.

**Stochastic Proximal Point Method.** The stochastic proximal point method (SPPM) is our starting point in developing SVRP and its Catalyzed variant. SPPM is well-studied, and we only briefly review some results on its convergence: Bertsekas (2011) terms it the incremental proximal point method, and provides analysis showing nonasymptotic convergence around a solution under the assumptions

that each $f_m$ is Lipschitz. Ryu & Boyd (2014) provide convergence rates for the algorithm, and observe that it is stable to learning rate misspecification unlike stochastic gradient descent (SGD). Pătraşcu & Necoara (2017) analyze SPPM for constrained optimization with random projections. Asi & Duchi (2019) study a more general method (AProx) that includes SPPM as a special case, giving stability and convergence rates under convexity. Asi et al. (2020); Chadha et al. (2022) further consider minibatching and convergence under interpolation for AProx.

In the context of federated learning for non-convex optimization, SPPM is also known as FedProx (Li et al., 2020b) and has been analyzed in several settings, see e.g. (Yuan & Li, 2022). Unfortunately, in the non-convex setting the convergence rates achieved by FedProx/SPPM are no better than SGD. SPPM can be applied to more than just federated learning and has found applications in matrix and tensor completion (Bumin & Huang, 2021) and reinforcement learning (Asadi et al., 2021). It can also be implemented efficiently for various optimization problems (Shtoff, 2022).

**Compression for communication efficiency.** Client sampling is one way of achieving communication efficiency, but there are other ways to do that in federated learning, such as compressing the vectors exchanged between the server and the clients. Szlendak et al. (2022); Beznosikov & Gasnikov (2022) consider federated optimization with compression under Assumption 1 and obtain better convergence rates using specially-crafted compression operators. Because these techniques are orthogonal, exploring combinations of client sampling and compression may be a promising avenue for future work.

## 3  PRELIMINARIES

We say that a differentiable function $f$ is $\mu$-strongly convex (for $\mu > 0$) if for all $x, y \in \mathbb{R}^d$ we have $f(x) \geq f(y) + \langle \nabla f(y), x - y \rangle + \frac{\mu}{2} \|x - y\|^2$. We assume:

**Assumption 2.** *All the functions* $f_1, f_2, \ldots, f_M$ *in problem* (1) *are* $\mu$-*strongly convex. We also assume Problem* (1) *has a solution* $x_* \in \mathbb{R}^d$.

The assumption that every $f_m$ is strongly convex is common in the analysis of federated learning algorithms (Karimireddy et al., 2020b; Mishchenko et al., 2022b) and is often realized when each $f_m$ represents a convex empirical loss with $\ell_2$-regularization. We assume that $f$ has a minimizer $x_*$, and by strong convexity this minimizer is unique. The proximal mapping associated with a function $h$ and stepsize $\eta > 0$ is defined as

$$\mathrm{prox}_{\eta h}(x) = \arg\min_{y \in \mathbb{R}^d} \left[ \eta h(y) + \tfrac{1}{2} \|y - x\|^2 \right].$$

When $h$ is convex, the minimization problem has a unique solution and hence the proximal operator is well-defined. We say that a point $y \in \mathbb{R}^d$ is a $b$-approximation of the proximal operator evaluated at $x$ if $\left\| y - \mathrm{prox}_{\eta h}(x) \right\|^2 \leq b$. When $h = f_m$ for some $m \in [M]$, computing the proximal operator is equivalent to solving a local optimization problem on node $m$.

## 4  ALGORITHMS AND THEORY

In this section we develop the main algorithms of our work for solving Problem 1 under Assumption 1, smoothness, and strong convexity. In the first subsection, we analyze the stochastic proximal point method and explore some of its desirable properties. Next, we augment the stochastic proximal point method with variance-reduction and develop SVRP, a novel algorithm that improves upon existing algorithms using client sampling. Finally, we use the Catalyst acceleration framework (Lin et al., 2015) to improve the convergence rate of SVRP. We give more details on how the algorithms are applied in a client-server setting in Appendix I.

### 4.1  BASICS: STOCHASTIC PROXIMAL POINT METHOD

The starting point of our investigation is the stochastic proximal point method (SPPM) (Algorithm 1), because the stochastic proximal point algorithm can achieve rates of convergence that are *smoothness-independent* and which rely only on the strong convexity of the minimization objectives (Asi &

---
**Algorithm 1:** Stochastic Proximal Point Method (SPPM)

---
**Data:** Stepsize $\eta$, initialization $x_0$, number of steps $K$, proximal solution accuracy $b$.

1 **for** $k = 0, 1, 2, \ldots, K - 1$ **do**

2     Sample $\xi_k \sim \mathcal{D}$.

3     Update with $b$-approximation of the stochastic proximal point operator:

$$x_{k+1} \simeq \text{prox}_{\eta f_{\xi_k}} (x_k).$$

---

[Duchi](), [2019]). This makes SPPM a much better starting point for developing new algorithms if our goal is to obtain convergence rates that depend on the dissimilarity constant $\delta$ instead of the (typically larger) smoothness constant $L$. The stochastic proximal point can be applied to the general *stochastic expectation* problem, which has the following form:

$$\min_{x \in \mathbb{R}^d} \left[ f(x) = \mathbb{E}_{\xi \sim \mathcal{D}} \left[ f_\xi(x) \right] \right], \tag{2}$$

where each $f_\xi$ is $\mu$-strongly convex and differentiable. Observe that Problem (1) is a special case of (2) where $\mathcal{D}$ has finite support. We assume that $f$ has a (necessarily unique) minimizer $x_*$. In this formulation, sampling a new $\xi \sim \mathcal{D}$ corresponds to sampling a node/client in federated optimization, and then a proximal iteration corresponds to a local optimization problem to be solved on node $\xi$. The next theorem characterizes the convergence of SPPM in this setting:

**Theorem 1.** *Suppose that $f_\xi$ is almost surely $\mu$-strongly convex, let $x_*$ be the minimizer of $f$, and define $\sigma_*^2 = \mathbb{E}_{\xi \sim \mathcal{D}} \left[ \|\nabla f_\xi(x_*)\|^2 \right]$. Suppose that for each $k$ we have that $x_{k+1}$ is a $b$-approximation of the proximal. Set $\eta = \frac{\mu \epsilon}{2\sigma_*^2}$ and $b \leq \frac{\epsilon}{4} \frac{(\eta \mu)^2}{(1 + \eta \mu)^2}$. Then $\mathbb{E} \left[ \|x_K - x_*\|^2 \right] \leq \epsilon$ after $K$ iterations:*

$$K = \left( 1 + \frac{2\sigma_*^2}{\mu^2 \epsilon} \right) \log \left( \frac{4\|x_0 - x_*\|^2}{\epsilon} \right). \tag{3}$$

The full proofs of all the theorems are relegated to the supplementary material. Theorem 1 for SPPM essentially gives the same rate as ([Asi & Duchi](), [2019], Proposition 5.3) but with a different proof technique. Our analysis relies on a straightforward application of the contractivity of the proximal, making it easier to extend to the variance-reduced case compared to the more involved analysis of ([Asi & Duchi](), [2019]).

**Comparison with SGD.** The iteration complexity of SGD in the same setting is:

$$K_{\text{SGD}} = \left( \frac{2L}{\mu} + \frac{2\sigma_*^2}{\mu^2 \epsilon} \right) \log \left( \frac{2\|x_0 - x_*\|^2}{\epsilon} \right). \tag{4}$$

See ([Needell et al.](), [2014]; [Gower et al.](), [2019]) for a derivation of this iteration complexity and ([Nguyen et al.](), [2019]) for a matching lower bound (up to log factors and constants). Observe that while the dependence on the stochastic noise term is the same in both (3) and (4), the iteration complexity of SGD also has an additional dependence on the condition number $\kappa = \frac{L}{\mu}$ while the iteration complexity of the stochastic proximal point method is entirely independent of the magnitude of the smoothness constant $L$. Thus, *we can obtain a faster convergence rate than SGD if we have access to stochastic proximal operator evaluations*. In federated optimization, a stochastic proximal operator evaluation can be done entirely with local work and with no communication, and thus is relatively cheap. Indeed, the iteration complexity of SPPM can even beat *accelerated* SGD because acceleration only reduces the dependence on the condition number from $L/\mu$ to $\sqrt{L/\mu}$, whereas SPPM has no dependence on the condition number to begin with.

**Communication vs computation complexities.** Every iteration of SPPM involves two communication steps: the server sends the current iterate $x_k$ to node $\xi$, and then node $\xi$ sends $x_{k+1}$ back to the server. Thus the communication complexity of SPPM is the same as eq. (3) multiplied by two. Each node needs to solve the optimization problem

$$\min_{x \in \mathbb{R}^d} \left[ f_\xi(x) + \frac{1}{2\eta} \|x - x_k\|^2 \right]$$

up to the accuracy $b$ given in Theorem 1. If each $f_\xi$ is $L$-smooth and $\mu$-strongly convex, this is a $L + \frac{1}{2\eta}$-smooth and $\mu + \frac{1}{2\eta}$-strongly convex minimization problem, and thus can be solved to the

---

**Algorithm 2:** Stochastic Variance-Reduced Proximal Point (SVRP) Method

**Data:** Stepsize $\eta$, initialization $x_0$, number of steps $K$, communication probability $p$, local
 solution accuracy $b$.

1 Initialize $w_0 = x_0$.
2 **for** $k = 0, 1, 2, \ldots, K - 1$ **do**
3  Sample $m_k$ uniformly at random from $[M]$.
4  Set
$$g_k = \nabla f(w_k) - \nabla f_{m_k}(w_k).$$
5  Compute a $b$-approximation of the stochastic proximal point operator associated with $f_{m_k}$:
$$x_{k+1} \simeq \operatorname{prox}_{\eta f_{m_k}} (x_k - \eta g_k). \tag{5}$$
6  Sample $c_k \sim \operatorname{Bernoulli}(p)$ and update $w_{k+1} = \begin{cases} x_{k+1} & \text{if } c_k = 1, \\ w_k & \text{if } c_k = 0. \end{cases}$

---

desired precision $b$ in $\mathcal{O}\left(\sqrt{\frac{\eta L+1}{\eta \mu+1}} \log \frac{1}{b}\right)$ local gradient accesses using accelerated gradient descent (Nesterov, 2018). When $\eta = \frac{\mu\epsilon}{4\sigma_*^2}$, this corresponds to a per-iteration computational complexity of $\mathcal{O}\left(\sqrt{\frac{\mu L\epsilon + 4\sigma_*^2}{\mu^2\epsilon + 4\sigma_*^2}} \log \frac{1}{b}\right)$. Note that if $f_\xi$ itself represents the loss on a local dataset (as is common in federated learning), we may use methods tailored for stochastic or finite-sum problems such as Random Reshuffling (Mishchenko et al., 2020), Katyusha (Allen-Zhu, 2017), or (accelerated) SVRG. Thus we see that, compared to SGD, *SPPM trades off a higher computational complexity for a lower communication complexity*.

**Related work.** We compare against related convergence results for SPPM in Appendix E.

## 4.2 THE SVRP ALGORITHM

The rate of SPPM, while independent of the condition number $\kappa = \frac{L}{\mu}$, is sublinear. While a sublinear rate is optimal for the stochastic oracle (Foster et al., 2019; Woodworth & Srebro, 2021), it is suboptimal in the setting of smooth finite-sum minimization (Woodworth & Srebro, 2016). In this section, we develop a novel variance-reduced method, SVRP (Stochastic Variance-Reduced Proximal Method, Algorithm 2), that converges linearly and relies only on second-order similarity.

Variance-reduced methods such as SVRG (Johnson & Zhang, 2013), SAGA (Defazio et al., 2014) or SARAH (Nguyen et al., 2017) improve the convergence rate of SGD for finite-sum problems by constructing gradient estimators whose variance vanishes over time. While SGD is used as the building block in most existing variance-reduced methods, in the preceding section we saw that the stochastic proximal point method is more communication-efficient; It stands to reason that variance-reduced variants of SPPM could also be more communication-efficient under second-order similarity. We apply variance-reduction to SPPM and develop our algorithm in the next two steps.

**Step (a): Adapting SVRG-style variance reduction to SPPM.** We use SVRG-style variance-reduction coupled with the stochastic proximal point method as a base. To see how, we start with SGD iterations: at each step $k$ we sample node $m_k$ uniformly at random from $[M]$ and update as
$$x_{k+1} = x_k - \eta \nabla f_{m_k}(x_k).$$

The main problem with SGD is that the stochastic gradient estimator has a non-vanishing variance that slows down convergence. SVRG (Johnson & Zhang, 2013) modifies SGD by adding a correction term $g_k$ at each iteration:
$$g_k = \nabla f(w_k) - \nabla f_{m_k}(w_k),$$
$$x_{k+1} = x_k - \eta \left[\nabla f_{m_k}(x_k) + g_k\right],$$

where $w_k$ is an anchor point that is periodically reset to the current iterate. Thus we added the correction term $g_k$ in order to reduce the variance in the gradient estimator and allow the algorithm to

converge. We propose to do the same for the stochastic proximal point method, where we instead change the argument to the proximal operator:

$$g_k = \nabla f(w_k) - \nabla f_{m_k}(w_k),$$
$$x_{k+1} = \text{prox}_{\eta f_{m_k}} \left( x_k - \eta g_k \right).$$

We can expect the correction term $g_k$ to function similarly and allow SPPM to converge faster.

**Step (b): Removing the loop.** Rather than reset the anchor point $w_k$ to the current iterate at fixed intervals, SVRP is instead *loopless*: it uses a random coin flip to determine when to communicate and re-compute full gradients. Loopless variants of variance-reduced algorithms such as L-SVRG (Kovalev et al., 2020) and Loopless SARAH (Li et al., 2020a) enjoy the same convergence guarantees as their ordinary counterparts, but with superior empirical performance and simpler analysis.

Combining the previous two steps gives us SVRP (Algorithm 2), and we give the convergence result next.

**Theorem 2.** *(Convergence of SVRP). Suppose that Assumptions 1 and 2 hold, and that each $x_{k+1}$ is a b-approximation of the proximal (5). Let $\tau = \min \left\{ \frac{\eta\mu}{1+2\eta\mu}, \frac{p}{2} \right\}$. Set the parameters of Algorithm 2 as $\eta = \frac{\mu}{2\delta^2}$, $b \leq \frac{\epsilon\tau(\eta\mu)^2}{2(1+\eta\mu)^3}$, and $p = \frac{1}{M}$. Then the final iterate $x_K$ satisfies $\mathbb{E}\left[ \|x_K - x_*\|^2 \right] \leq \epsilon$ provided that the total number of iterations $K$ is larger than $T_{\text{iter}}$:*

$$T_{\text{iter}} = \tilde{\mathcal{O}} \left( \left( M + \frac{\delta^2}{\mu^2} \right) \log \frac{1}{\epsilon} \right).$$

**Communication complexity.** We consider one communication to represent the server exchanging one vector with one client. At each step of SVRP, the server samples a client $m_k$ and sends them the current iterate $x_k$, the client then computes $g_k$ and $x_{k+1}$ locally, and sends $x_{k+1}$ back to the server. Then, with probability $p$, the server changes the anchor point $w_{k+1}$ to $x_{k+1}$, sends $w_{k+1}$ to the new clients, each client $m$ then computes $\nabla f_m(w_{k+1})$ and sends it back to the server, which averages the received gradients to get $\nabla f(w_{k+1})$; The server then proceeds to send $\nabla f(w_{k+1})$ back to all the clients. Thus the expected communication complexity is

$$\mathbb{E}\left[ T_{\text{comm}} \right] = (2 + 3pM)\, T_{\text{iter}} = 5 T_{\text{iter}} = \tilde{\mathcal{O}} \left( \left( M + \frac{\delta^2}{\mu^2} \right) \log \frac{1}{\epsilon} \right).$$

Compared to the SVRG communication complexity $\tilde{\mathcal{O}} \left( \left( M + \frac{L}{\mu} \right) \log \frac{1}{\epsilon} \right)$ (Sebbouh et al., 2019), this replaces the $L/\mu$ dependence with a $\delta^2/\mu^2$ dependence. This is better when $\delta \leq \sqrt{L\mu}$.

**Comparison with existing results.** The lower bound related the most to our setting is given by (Arjevani & Shamir, 2015), only considers the setting of full participation (i.e. no client sampling) and corresponds to a communication complexity of $\tilde{\mathcal{O}} \left( \sqrt{\frac{\delta}{\mu}} M \right)$. Our result improves upon this when $M > (\delta/\mu)^{3/2}$. Note that while our result attains a superior dependence on $M$, the dependence on the effective conditioning $\delta/\mu$ is worse. In the next section, we shall improve this via acceleration. Note that the best existing results for optimization under second-order similarity, such as DiSCO (Zhang & Lin, 2015), Accelerated SONATA (Tian et al., 2022), and Extragradient sliding (Kovalev et al., 2022), match this lower bound (Kovalev et al., 2022, Table 1). Therefore, our result shows that significantly better convergence can be obtained under second-order similarity when using client sampling.

**Computational complexity**. Similar to the discussion of the computational complexity of SPPM in the previous section, here too we essentially need to solve on each sampled device an optimization problem involving an $(L+\eta)$-smooth and $(\mu+\eta)$-strongly convex function up to the accuracy $b$. This can be done using accelerated gradient descent in $\tilde{\mathcal{O}} \left( \sqrt{\frac{L+\eta}{\mu+\eta}} \log \frac{1}{b} \right) = \tilde{\mathcal{O}} \left( \sqrt{\frac{\frac{L\delta^2}{\mu}+1}{2\delta^2+1}} \log \frac{1}{b} \right)$ gradient accesses. Compared to SGD or SVRG, we can see that we trade off a higher local Computational complexity for a lower number of communications steps to convergence.

**Similar methods.** Point-SAGA (Defazio, 2016) uses the stochastic proximal point method coupled with SAGA-style variance reduction. Point-SAGA achieves optimal performance under smoothness

and strong convexity, but it inherits the heavy memory requirements of SAGA and its performance under Assumption 1 is unknown. Another similar algorithm is SCAFFOLD (Karimireddy et al., 2020b): SCAFFOLD uses a similar SVRG-style correction sequence, and their method can be viewed as approximately solving a local unregularized minimization problem at each step. Unfortunately, their analysis under Assumption 1 only holds for quadratic objectives and without client sampling.

## 4.3 ACCELERATING SVRP VIA CATALYST

In this section we improve SVRP by augmenting it with acceleration. Acceleration is an effective way to improve the convergence rate of first-order gradient methods, capable of achieving better convergence rates in deterministic (Nesterov, 2018), finite-sum (Allen-Zhu, 2017), and stochastic (Jain et al., 2018) settings. Catalyst (Lin et al., 2015) is a generic framework for accelerated first-order methods that we utilize to accelerate SVRP. Catalyst (Algorithm 3) is essentially an accelerated proximal point method that relies on a solver $\mathcal{A}$ to solve the proximal point iterations. We use SVRP (Algorithm 2) as the solver $\mathcal{A}$, and term the resulting algorithm *Catalyzed SVRP*.

Catalyst gives us fast linear convergence provided that we solve certain regularized subproblems to a high accuracy. To do that, we run SVRP (as the method $\mathcal{A}$) with an appropriate set of parameters and for a fixed number of iterations. The details of our application of Catalyst are given in the supplementary material (Appendix G). The complexity of Catalyzed SVRP is given next.

**Theorem 3.** *(Catalyzed SVRP convergence rate). For Catalyst (Algorithm 3) with SVRP (Algorithm 2) as the inner solver $\mathcal{A}$, suppose Assumptions 1 and 2 hold, and that $f$ is $L$-smooth. Then for a specific choice of the algorithm parameters, the expected communication complexity $\mathbb{E}\left[\mathrm{T}_{\mathrm{comm}}\right]$ to reach an accuracy $\epsilon$, up to polylogarithmic factors and absolute constants, is*

$$\mathbb{E}\left[\mathrm{T}_{\mathrm{comm}}\right] = \tilde{\mathcal{O}}\left(\left(M + \sqrt{\tfrac{\delta}{\mu}}M^{3/4}\right)\log\tfrac{1}{\epsilon}\right). \tag{6}$$

**Improvement over SVRP.** Compared to ordinary SVRP, observe that since $\sqrt{\tfrac{\delta}{\mu}}M^{3/4} \leq M + (\delta/\mu)^2$, Theorem 3 gives a communication complexity that is always better than what Theorem 2 provides up to logarithmic factors. In particular, the rate of Catalyzed SVRP given by (6) is strictly better than the rate of vanilla SVRP when $\tfrac{\delta}{\mu} \leq \sqrt{M}$. Note that unlike the communication complexity given by Theorem 2, the rate given by Theorem 3 requires smoothness, but the smoothness constant $L$ only shows up inside a logarithmic factor, and hence does not show up in eq. (6) (as $\tilde{\mathcal{O}}$ notation hides polylogarithmic factors).

**Comparison with the smooth setting.** When each function is $L$-smooth and $\mu$-strongly convex, Accelerated variants of SVRG reach an $\epsilon$-accurate solution in $\tilde{\mathcal{O}}\left(M + \sqrt{\tfrac{L}{\mu}}M^{1/2}\right)$ communication steps and $\mathcal{O}(1)$ local work per node (Lin et al., 2015). The communication complexity Catalyzed SVRP achieves is better when $\delta \leq \tfrac{L}{\sqrt{M}}$, at the cost of more local computation.

**Improvement over prior work.** The best existing algorithms for solving Problem (1) under Assumption 1 are Accelerated SONATA (Tian et al., 2022) and Extragradient sliding (Kovalev et al., 2022), both of which achieve a communication complexity of $\tilde{\mathcal{O}}\left(\sqrt{\tfrac{\delta}{\mu}}M\right)$. The rate given by eq. (6) is better than this rate, achieving a smaller dependence on the number of nodes $M$. Arjevani & Shamir (2015) give a lower bound matching the rate $\tilde{\mathcal{O}}\left(\sqrt{\tfrac{\delta}{\mu}}M\right)$: ignoring polylogarithmic factors, our result improves upon their lower bound through the usage of client sampling. This improvement is possible because Arjevani & Shamir (2015) use an oracle model that does not allow for client sampling. Thus, *Catalyzed SVRP improves upon all existing methods for optimization under Assumption 1, smoothness, and strong convexity when $\delta$ is sufficiently small.*

**Computational complexity.** Catalyzed SVRP uses SVRP as an inner solver, and hence inherits its computational requirements: at each iteration $k$, we have to sample a node $m_k$ and evaluate the proximal operator associated with its local objective $f_{m_k}$. However, we set the solution accuracy to $b = 0$ when applying Catalyst to SVRP, i.e. we require exact stochastic proximal operator evaluation; This is only done for convenience of analysis, and we believe the analysis can be extended to the case of nonzero but small $b$. In practice, the proximal operation can be computed exactly in many cases,

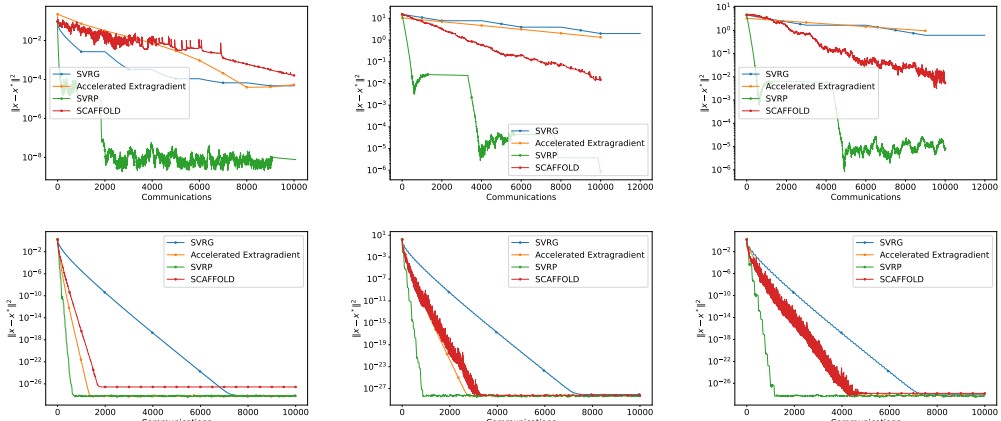

Figure 1: Top row: experiments on synthetic data. Left figure has $M = 1000$ clients, middle has $M = 2000$, and right has $M = 3000$. Bottom row: experiments on real data. Left has $M = 20$ clients, middle has $M = 40$ clients, and right has $M = 60$ clients. We plot the squared distance from the optimum point for all methods on a logarithmic scale versus the number of communication steps, where one exchange of vectors between the server and a single client counts as a communication step.

and otherwise we can compute the proximal operator to machine accuracy using accelerated gradient descent.

## 5 EXPERIMENTS

We run linear regression with $\ell_2$ regularization, where each client has a loss function of the form

$$f(x) = \frac{1}{M} \sum_{m=1}^{M} \left[ f_m(x) = \frac{1}{n} \sum_{i=1}^{n} (z_{m,i}^\top x - y_{m,i})^2 + \frac{\lambda}{2} \|x\|^2 \right]$$

where $z_{m,i} \in \mathbb{R}^d$ and $y_{m,i} \in \mathbb{R}$ represent the feature and label vectors respectively, for $m \in [M]$ and $i \in [n]$. We do two sets of experiments: in the first set, we generate the data vectors $z_{m,i}$ synthetically and force the second-order similarity assumption to hold with $\delta$ small relative to $L$, with $L \approx 3330$ and $\delta \approx 10$, and regularization constant $\lambda = 1$. We vary the number of clients $M$ as $M \in \{1000, 2000, 3000\}$. In the second set, we use the "a9a" dataset from LIBSVM (Chang & Lin, 2011), each client's data is constructed by sampling from the original training dataset with $n = 2000$ samples per client. We vary the number of clients $M$ as $M \in \{20, 40, 60\}$, and measure $L \approx 6.33$, $\delta \approx 0.22$, and set the regularization parameter as $\lambda = 0.1$. We simulate our results on a single machine, running each method for 10000 communication steps. Our results are given in Figure 1. We compare SVRP against SVRG, SCAFFOLD, and the Accelerated Extragradient algorithms, using the optimal theoretical stepsize for each algorithm. In all cases, SVRP achieves superior performance to existing algorithms for both the real and synthetic data experiments. This is more pronounced in the synthetic data experiments as $\delta$ is much smaller than $L$ and the number of agents $M$ is very large.

## 6 CONCLUSION

In this paper, we develop two algorithms that utilize client sampling in order to reduce the amount of communication necessary for federated optimization under the second-order similarity assumption, with the faster of the two algorithms reducing the best-known communication complexity from $\tilde{\mathcal{O}}\left(\sqrt{\frac{\delta}{\mu}}M\right)$ (Kovalev et al., 2022) to $\tilde{\mathcal{O}}\left(\sqrt{\frac{\delta}{\mu}}M^{3/4} + M\right)$. Both algorithms utilize variance-reduction on top of the stochastic proximal point algorithm, for which we also provide a new simple and smoothness-free analysis.

In all cases, the algorithms tradeoff more local work for a reduced communication complexity. An important direction in future research is to investigate whether this tradeoff is necessary, and to derive lower bounds under the partial participation communication model and second-order similarity.

## REPRODUCIBILITY STATEMENT

All the theoretical results we derive are accompanied by full proofs in the appendices, and all the definitions and assumptions we make are clearly stated in the main text. We attach the code used to run the experiments as supplementary material to the paper.

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

CONTENTS

## A    BASIC FACTS AND NOTATION

We shall make use of the following facts from linear algebra: for any $a, b \in \mathbb{R}^d$ and any $\zeta > 0$,

$$2 \langle a, b \rangle = \|a\|^2 + \|b\|^2 - \|a - b\|^2. \tag{7}$$

$$\|a\|^2 \leq (1 + \zeta) \|a - b\|^2 + \left(1 + \zeta^{-1}\right) \|b\|^2. \tag{8}$$

# B  APPLICATIONS OF SECOND-ORDER SIMILARITY

In this section we give a few examples where Assumption 1 holds. These are known in the literature, and we collect them here for motivation.

**Relation to smoothness.**  The standard assumption in analyzing federated optimization algorithms for Problem 1 is smoothness:

**Definition 4.** (Smoothness) We say that a differentiable function $f$ is $L$-smooth if for all $x, y \in \mathbb{R}^d$ we have $\|\nabla f(x) - \nabla f(y)\| \leq L \|x - y\|$.

Note that $L$-smoothness of each $f_1, f_2, \ldots, f_M$ implies Assumption 1 holds with $\delta = L$, as we have for any $x, y \in \mathbb{R}^d$

$$\frac{1}{M} \sum_{m=1}^{M} \|\nabla f_m(x) - \nabla f(x) - [\nabla f_m(y) - \nabla f(y)]\|^2$$

$$= \frac{1}{M} \sum_{m=1}^{M} \|\nabla f_m(x) - \nabla f_m(y)\|^2 - \|\nabla f(x) - \nabla f(y)\|^2$$

$$\leq \frac{1}{M} \sum_{m=1}^{M} \|\nabla f_m(x) - \nabla f_m(y)\|^2$$

$$\leq L^2 \|x - y\|^2.$$

Thus we have that, under smoothness, Assumption 1 holds. While typically we are interested in the case in which $\delta$ is much smaller than $L$, the fact that by default we have $\delta \leq L$ means that we do not lose any generality by considering optimization under Assumption 1.

**Relation to mean-squared smoothness.**  Observe that in the preceding proof we did not actually use that each $f_m$ is $L$-smooth, only that

$$\frac{1}{M} \sum_{m=1}^{M} \|\nabla f_m(x) - \nabla f_m(y)\|^2 \leq L^2 \|x - y\|^2.$$

This assumption is known as *mean-squared smoothness* in the literature, and this proof shows it is also a special case of second-order similarity.

**Statistical Learning.**  Suppose that each function $f_m$ corresponds to empirical risk minimization with data drawn according to a distribution $\mathcal{D}_w$:

$$f_m(x) = \frac{1}{n} \sum_{i=1}^{n} \ell(x, z_{m,i}), \tag{9}$$

where $\ell$ is $L$-smooth and convex in its first argument and $z_{m,i}$ represents the $i$-th training point on node $m$. If all the training points are drawn i.i.d. from the same distribution $z_{m,i} \sim \mathcal{Z}$ for all $m \in [M]$ and $i \in [n]$, then if the losses are quadratic, Shamir et al. (2014) show that, with high probability, Assumption 1 holds with $\delta = \tilde{\mathcal{O}}\left(\frac{L}{\sqrt{n}}\right)$. This can be much smaller than $L$, especially when the number of data points per node $n$ is large. Zhang & Lin (2015) show that a similar concentration holds for non-quadratic minimization, albeit with extra dependence on the data dimensionality $d$. Hendrikx et al. (2020) remove the dependence on the data dimensionality for generalized linear models under some additional assumptions.

While clients normally do not ordinarily share the same data distribution $\mathcal{D}_w$ in federated optimization, *clustering* clients together such that each group of clients has similar data is a common strategy (Sattler et al., 2020; Ghosh et al., 2020), and as such we can apply algorithms designed for Assumption 1 to clusters of clients with similar data.

**Other examples.**  Karimireddy et al. (2020b) observe that Assumption 1 holds with $\delta = 0$ when using objective perturbation as a differential privacy mechanism for empirical risk minimization, since objective perturbation relies on adding linear noise that does not affect the differences of gradients. Chayti & Karimireddy (2022) give more examples relevant to federated learning.

**Hessian formulation.** The way we wrote Assumption 1 in the main text is

$$\frac{1}{M} \sum_{m=1}^{M} \|\nabla f_m(x) - \nabla f(x) - [\nabla f_m(y) - \nabla f(y)]\|^2 \leq \delta^2 \|x - y\|^2.$$

For twice-differentiable objectives, this is a biproduct of the following inequality on the Hessians: for all $x \in \mathbb{R}^d$ we have

$$\frac{1}{M} \sum_{m=1}^{M} \|\nabla^2 f_m(x) - \nabla^2 f(x)\|_{op}^2 \leq \delta^2.$$

This motivates the name *second-order similarity*. To see why this is the case, observe that by Taylor's theorem (Duistermaat, 2004, Theorem 2.8.3), Jensen's inequality and the convexity of the squared norm, we have

$$\|\nabla f_m(x) - \nabla f(x) - [\nabla f_m(y) - \nabla f(y)]\|^2$$

$$= \left\| \int_0^1 \left[ \nabla^2 f_m(\theta x + (1-\theta)y) - \nabla^2 f(\theta x + (1-\theta)y) \right] (x-y) \mathrm{d}\theta \right\|^2$$

$$\leq \int_0^1 \left\| \left[ \nabla^2 f_m(\theta x + (1-\theta)y) - \nabla^2 f(\theta x + (1-\theta)y) \right] (x-y) \right\|^2 \mathrm{d}\theta$$

$$\leq \int_0^1 \left\| \nabla^2 f_m(\theta x + (1-\theta)y) - \nabla^2 f(\theta x + (1-\theta)y) \right\|_{op}^2 \|x-y\|^2 \mathrm{d}\theta.$$

Averaging with respect to $m$ gives

$$\frac{1}{M} \sum_{m=1}^{M} \|\nabla f_m(x) - \nabla f(x) - [\nabla f_m(y) - \nabla f(y)]\|^2$$

$$\leq \int_0^1 \frac{1}{M} \sum_{m=1}^{M} \left\| \nabla^2 f_m(\theta x + (1-\theta)y) - \nabla^2 f(\theta x + (1-\theta)y) \right\|_{op}^2 \|x-y\|^2 \mathrm{d}\theta$$

$$\leq \int_0^1 \delta^2 \|x-y\|^2 \mathrm{d}\theta$$

$$= \delta^2 \|x-y\|^2.$$

Therefore if it holds that $\max_{m \in [M]} \sup_{x \in \mathbb{R}^d} \left\| \nabla^2 f_m(x) - \nabla f(x) \right\|_{op} \leq \delta$ (a condition known as Hessian similarity), we necessarily have that Assumption 1 also holds.

## C ALGORITHM-INDEPENDENT RESULTS

This section collects all facts and propositions that are algorithm-independent.

### C.1 FACTS ABOUT THE PROXIMAL OPERATOR

In this section we derive two useful facts about the proximal operator. Both facts are relatively straightforward to derive.

**Fact 1.** *Let $h : \mathbb{R}^d \to \mathbb{R}$ be a convex differentiable function and $\eta > 0$ be a stepsize. Then for all $x \in \mathbb{R}^d$,*

$$\mathrm{prox}_{\eta h}(x + \eta \nabla h(x)) = x.$$

*Proof.* Solving the proximal is equivalent to

$$\mathrm{prox}_{\eta h}(z) = \operatorname*{argmin}_{y \in \mathbb{R}^d} \left( h(y) + \frac{1}{2\eta} \|y - z\|^2 \right).$$

This is a strongly convex minimization problem for any $\eta > 0$, hence the (necessarily unique) minimizer of this problem satisfies the first order optimality condition

$$\nabla h(y) + \frac{1}{\eta}(y - z) = 0.$$

Now observe that we have

$$\nabla h(x) + \frac{1}{\eta}(x - (x + \eta \nabla h(x))) = \nabla h(x) + \frac{-\eta \nabla h(x)}{\eta} = 0.$$

It follows that $x = \text{prox}_{\eta h}(x + \eta \nabla h(x))$. ∎

**Fact 2.** *(Tight contractivity of the proximal operator). If $h$ is $\mu$-strongly convex and differentiable, then for all $\eta > 0$ and for any $x, y \in \mathbb{R}^d$ we have*

$$\left\| \text{prox}_{\eta h}(x) - \text{prox}_{\eta h}(y) \right\|^2 \leq \frac{1}{(1 + \eta \mu)^2} \|x - y\|^2$$

*Proof.* This lemma can be seen as a tighter version of (Mishchenko et al., 2022a, Lemma 5) though our proof technique is different. Note that $p(x) = \text{prox}_{\eta h}(x)$ satisfies $\eta \nabla h(p(x)) + [p(x) - x] = 0$, or equivalently $p(x) = x - \eta \nabla h(p(x))$. Using this we have

$$
\begin{aligned}
\|p(x) - p(y)\|^2 &= \|[x - \eta \nabla h(p(x))] - [y - \eta \nabla h(p(y))]\|^2 \\
&= \|[x - y] - \eta [\nabla h(p(x)) - \nabla h(p(y))]\|^2 \\
&= \|x - y\|^2 + \eta^2 \|\nabla h(p(x)) - \nabla h(p(y))\|^2 - 2\eta \langle x - y, \nabla h(p(x)) - \nabla h(p(y)) \rangle.
\end{aligned}
$$
(10)

Now note that

$$
\begin{aligned}
\langle x - y, \nabla h(p(x)) - \nabla h(p(y)) \rangle &= \langle p(x) + \eta \nabla h(p(x)) - [p(y) + \eta \nabla h(p(y))], \nabla h(p(x)) - \nabla h(p(y)) \rangle \\
&= \langle p(x) - p(y), \nabla h(p(x)) - \nabla h(p(y)) \rangle + \eta \|\nabla h(p(x)) - \nabla h(p(y))\|^2.
\end{aligned}
$$
(11)

Combining eqs. (10) and (11) we get

$$
\begin{aligned}
\|p(x) - p(y)\|^2 &= \|x - y\|^2 + \eta^2 \|\nabla h(p(x)) - \nabla h(p(y))\|^2 - 2\eta \langle p(x) - p(y), \nabla h(p(x)) - \nabla h(p(y)) \rangle \\
&\quad - 2\eta^2 \|\nabla h(p(x)) - \nabla h(p(y))\|^2 \\
&= \|x - y\|^2 - \eta^2 \|\nabla h(p(x)) - \nabla h(p(y))\|^2 - 2\eta \langle p(x) - p(y), \nabla h(p(x)) - \nabla h(p(y)) \rangle.
\end{aligned}
$$
(12)

Let $D_h(u, v) = h(u) - h(v) - \langle \nabla h(v), u - v \rangle$ be the Bregman divergence associated with $h$ at $u, v$. It is easy to show that

$$\langle u - v, \nabla h(u) - \nabla h(v) \rangle = D_h(u, v) + D_h(v, u).$$

This is a special case of the three-point identity (Chen & Teboulle, 1993, Lemma 3.1). Using this with $u = p(x)$ and $v = p(y)$ and plugging back into (12) we get

$$\|p(x) - p(y)\|^2 = \|x - y\|^2 - \eta^2 \|\nabla h(p(x)) - \nabla h(p(y))\|^2 - 2\eta [D_h(p(x), p(y)) + D_h(p(y), p(x))].$$

Note that because $h$ is strongly convex, we have that $D_h(p(y), p(x)) \geq \frac{\mu}{2} \|p(y) - p(x)\|^2$ and $D_h(p(x), p(y)) \geq \frac{\mu}{2} \|p(y) - p(x)\|^2$, hence

$$\|p(x) - p(y)\|^2 \leq \|x - y\|^2 - \eta^2 \|\nabla h(p(x)) - \nabla h(p(y))\|^2 - 2\eta \mu \|p(x) - p(y)\|^2.$$
(13)

Strong convexity implies that for any two points $u, v$

$$\|\nabla h(u) - \nabla h(v)\|^2 \geq \mu^2 \|u - v\|^2,$$

see (Nesterov, 2018, Theorem 2.1.10) for a proof. Using this in eq. (13) with $u = p(x)$ and $v = p(y)$ yields

$$\|p(x) - p(y)\|^2 \leq \|x - y\|^2 - \eta^2 \mu^2 \|p(x) - p(y)\|^2 - 2\eta \mu \|p(x) - p(y)\|^2.$$

Rearranging gives

$$\left[1 + \eta^2 \mu^2 + 2\eta \mu\right] \|p(x) - p(y)\|^2 \leq \|x - y\|^2.$$

It remains to notice that $(1 + \eta \mu)^2 = 1 + \eta^2 \mu^2 + 2\eta \mu$. ∎

## C.2 A LEMMA FOR SOLVING RECURRENCES

**Lemma 1.** *Suppose that we have a sequence of positive values $(r_k)_{k=0}^{K-1}$ satisfying, for some $\theta > 0$ and some $c > 0$*

$$r_{k+1} \le \frac{1}{1+\theta}\left[r_k + c\right].$$

*Then the sequence satisfies*

$$r_K \le \frac{1}{(1+\theta)^K}r_0 + \min\left\{\frac{K}{1+\theta}, \frac{1}{\theta}\right\}c.$$

*Proof.* We start from the recurrence to get

$$
\begin{aligned}
r_{k+1} &\le \frac{1}{1+\theta}r_k + \frac{c}{1+\theta} \\
&\le \frac{1}{1+\theta}\left[\frac{1}{1+\theta}r_{k-1} + \frac{c}{1+\theta}\right] + \frac{c}{1+\theta} \\
&= \frac{1}{(1+\theta)^2}r_{k-1} + \frac{c}{(1+\theta)^2} + \frac{c}{(1+\theta)}.
\end{aligned}
$$

Continuing similarly we obtain

$$r_K \le \frac{1}{(1+\theta)^K}r_0 + \frac{c}{1+\theta}\sum_{j=0}^{K-1}\left(\frac{1}{1+\theta}\right)^j. \tag{14}$$

We can now bound the latter sum in two ways, the first is to note that since $1 + \theta > 1$ we have that $\frac{1}{1+\theta} < 1$, and hence

$$\sum_{j=0}^{K-1}\left(\frac{1}{1+\theta}\right)^j \le \sum_{j=0}^{K-1} 1^j = K. \tag{15}$$

The second way is to use the convergence of the geometric series

$$\sum_{j=0}^{K-1}\left(\frac{1}{1+\theta}\right)^j \le \sum_{j=0}^{\infty}\left(\frac{1}{1+\theta}\right)^j = \frac{1}{1 - \frac{1}{1+\theta}} = \frac{1+\theta}{\theta}. \tag{16}$$

Using eqs. (15) and (16) in (14) gives

$$
\begin{aligned}
r_K &\le \frac{1}{(1+\theta)^K}r_0 + \frac{c}{1+\theta}\min\left\{K, \frac{1+\theta}{\theta}\right\} \\
&= \frac{1}{(1+\theta)^K}r_0 + c\cdot\min\left\{\frac{K}{1+\theta}, \frac{1}{\theta}\right\},
\end{aligned}
$$

and this is the lemma's statement. ■

## D  PROOFS FOR SPPM (ALGORITHM 1)

*Proof of Theorem 1.* Using eq. (8) and our assumption that the proximal operators are solved up to the accuracy $b$ we have

$$
\begin{aligned}
\|x_{k+1} - x_*\|^2 &= \left\|x_{k+1} - \mathrm{prox}_{\eta f_{\xi_k}}(x_k) + \mathrm{prox}_{\eta f_{\xi_k}}(x_k) - x_*\right\|^2 \\
&\le \left(1 + \frac{1}{\eta\mu}\right)\left\|x_{k+1} - \mathrm{prox}_{\eta f_{\xi_k}}(x_k)\right\|^2 + (1+\eta\mu)\left\|\mathrm{prox}_{\eta f_{\xi_k}}(x_k) - x_*\right\|^2 \\
&\le \left(\frac{1+\eta\mu}{\eta\mu}\right)b + (1+\eta\mu)\left\|\mathrm{prox}_{\eta f_{\xi_k}}(x_k) - x_*\right\|^2. \tag{17}
\end{aligned}
$$

For the second term in eq. (17) we have by Fact 1 that $x_* = \text{prox}_{\eta f_{\xi_k}}(x_* + \eta \nabla f_{\xi_k}(x_*))$, then using the contraction of the prox (Fact 2) we get

$$\left\| \text{prox}_{\eta f_{\xi_k}}(x_k) - x_* \right\|^2 = \left\| \text{prox}_{\eta f_{\xi_k}}(x_k) - \text{prox}_{\eta f_{\xi_k}}(x_* + \eta \nabla f_{\xi_k}(x_*)) \right\|^2$$
$$\leq \frac{1}{(1+\eta\mu)^2} \|x_k - (x_* + \eta \nabla f_{\xi_k}(x_*))\|^2. \qquad (18)$$

Expanding out the square we have

$$\left\| \text{prox}_{\eta f_{\xi_k}}(x_k) - x_* \right\|^2 \leq \frac{1}{(1+\eta\mu)^2} \|x_k - x_* - \eta \nabla f_{\xi_k}(x_*)\|^2$$
$$= \frac{1}{(1+\eta\mu)^2} \left[ \|x_k - x_*\|^2 + \eta^2 \|\nabla f_{\xi_k}(x_*)\|^2 - 2\eta \langle x_k - x_*, \nabla f_{\xi_k}(x_*) \rangle \right].$$

Denote expectation conditional on $x_k$ by $\mathbb{E}_k[\cdot]$. Taking expectation conditional on $x_k$ we get

$$\mathbb{E}_k \left[ \left\| \text{prox}_{\eta f_{\xi_k}}(x_k) - x_* \right\|^2 \right] \leq \frac{1}{(1+\eta\mu)^2} \left[ \|x_k - x_*\|^2 + \eta^2 \mathbb{E} \left[ \|\nabla f_{\xi_k}(x_*)\|^2 \right] - 2\eta \langle x_k - x_*, \mathbb{E}_k [\nabla f_{\xi_k}(x_*)] \rangle \right]$$
$$= \frac{1}{(1+\eta\mu)^2} \left[ \|x_k - x_*\|^2 + \eta^2 \sigma_*^2 \right],$$

where we used that $\mathbb{E}[\nabla f_{\xi_k}(x_*)] = \nabla f(x_*) = 0$ and the definition of $\sigma_*^2$. Taking conditional expectation in eq. (17) and plugging the last line gives

$$\mathbb{E}_k \left[ \|x_{k+1} - x_*\|^2 \right] \leq \left( \frac{1+\eta\mu}{\eta\mu} \right) b + \frac{1+\eta\mu}{(1+\eta\mu)^2} \left[ \|x_k - x_*\|^2 + \eta^2 \sigma_*^2 \right]$$
$$= \frac{1}{1+\eta\mu} \left[ \|x_k - x_*\|^2 + \eta^2 \sigma_*^2 + \frac{(1+\eta\mu)^2}{\eta\mu} b \right].$$

Taking unconditional expectation gives

$$\mathbb{E} \left[ \|x_{k+1} - x_*\|^2 \right] \leq \frac{1}{1+\eta\mu} \left[ \mathbb{E} \left[ \|x_k - x_*\|^2 \right] + \eta^2 \sigma_*^2 + \frac{(1+\eta\mu)^2}{\eta\mu} b \right].$$

We can then use Lemma 1 to get that at step $K$ we have

$$\mathbb{E} \left[ \|x_K - x_*\|^2 \right] \leq \left( \frac{1}{1+\eta\mu} \right)^K \|x_0 - x_*\|^2 + \frac{\eta\sigma_*^2}{\mu} + \frac{(1+\eta\mu)^2}{(\eta\mu)^2} b. \qquad (19)$$

This proves the first statement of the theorem. For the second statement, observe that when $\eta = \frac{\mu\epsilon}{2\sigma_*^2}$ and $b \leq \frac{\epsilon}{4} \frac{(\eta\mu)^2}{(1+\eta\mu)^2}$ we have

$$\frac{\eta\sigma_*^2}{\mu} + \frac{(1+\eta\mu)^2}{(\eta\mu)^2} b \leq \frac{\epsilon}{2} + \frac{\epsilon}{4} = \frac{3\epsilon}{4}. \qquad (20)$$

Moreover, we have using the inequality $1 - x \leq \exp(-x)$ for all $x > 0$ and our choice of $K$ that

$$\left( \frac{1}{1+\eta\mu} \right)^K = \left( 1 - \frac{\eta\mu}{1+\eta\mu} \right)^K \leq \exp \left( -\frac{\eta\mu K}{1+\eta\mu} \right) \leq \frac{\epsilon}{4} \frac{1}{\|x_0 - x_*\|^2}. \qquad (21)$$

Plugging eqs. (20) and (21) into eq. (19) yields

$$\mathbb{E} \left[ \|x_K - x_*\|^2 \right] \leq \epsilon,$$

and this is the second statement of the theorem. ∎

## E    RELATED WORK FOR SPPM

Toulis et al. (2015) study SPPM where they assume each stochastic proximal operation has unbiased and bounded error from the proximal operation with respect to the full proximal, i.e. for $\epsilon_\xi(x) = \frac{1}{\eta}\left[\text{prox}_{\eta f_\xi}(x) - \text{prox}_{\eta f}(x)\right]$ we have:

$$\mathbb{E}\left[\epsilon_\xi\right] = 0, \qquad\qquad \text{and,} \qquad\qquad \mathbb{E}\left[\|\epsilon_\xi\|^2\right] \leq \sigma^2. \qquad (22)$$

Under this assumption, Toulis et al. (2015) prove that the stochastic proximal point method can achieve a rate that is completely independent of the smoothness constant, and which matches the optimal rate for eq. (2) for (potentially) nonsmooth objectives. Kim et al. (2022) show a similar result under the same condition for a momentum variant of SPPM. It is not clear how to satisfy (22) in practice: the next example shows that even in the simple setting of quadratic minimization, the iterations are not unbiased.

**Example 1.** *Take $f_1 = ax^2$, $f_2 = 2ax^2$, and $f = \frac{1}{2}(f_1 + f_2)$, then for any $x \in \mathbb{R}^d$ we have for $\xi$ drawn uniformly at random from $\{1, 2\}$:*

$$\mathbb{E}\left[\text{prox}_{\eta f_\xi}(x) - \text{prox}_{\eta f}(x)\right] = \frac{(1 + 3\eta a)x}{1 + 6\eta a + 8\eta^2 a^2} - \frac{x}{3\eta a + 1} \neq 0.$$

*Thus the errors are* not *unbiased. Moreover, it can be shown that the variance scales with $\|x\|^2$, and thus can be made very large.*

In contrast, the convergence rate given by Theorem 1 does not require condition (22), and results in linear convergence for the functions of Example 1 (as both $f_1$ and $f_2$ share a minimizer at $x = 0$). We note that Ryu & Boyd (2014) also study the convergence of SPPM without a condition like (22), but their theory does not show convergence to an $\epsilon$-approximate solution even if the stepsize is taken proportional to $\epsilon$. Pătraşcu & Necoara (2017) also analyze the convergence of SPPM and present two different cases: if the stepsize is held constant (as in our Theorem 1), then their theory shows only convergence to a neighborhood whose size cannot be made small by varying the stepsize; Alternatively, by using a decreasing stepsize they can show a $\mathcal{O}\left(\frac{1}{\epsilon}\right)$ iteration complexity, but then this complexity requires that each $f_\xi$ is smooth. As mentioned previously, Asi & Duchi (2019, Proposition 5.3) show the same $\mathcal{O}\left(\frac{1}{\epsilon}\right)$ rate without requiring smoothness or a condition like (22), but using exact evaluations of the proximal operator. Theorem 1 gives the same iteration complexity without requiring smoothness or bounded variance, and while allowing for approximate proximal point operator evaluations. Note that the improvement of allowing approximate evaluations over (Asi & Duchi, 2019) is not very significant, as the approximation has to be very tight. The main way we depart from (Asi & Duchi, 2019) is that we use the contractivity of the proximal as the main proof tool, and this extends more easily to the variance-reduced setting of SVRP.

## F    PROOFS FOR SVRP (ALGORITHM 2)

*Proof of Theorem 2.* Let $\tilde{x}_{k+1} = \text{prox}_{\eta f_{m_k}}(x_k - \eta g_k)$. Then by eq. (8) and our assumption that $\|x_{k+1} - \tilde{x}_{k+1}\|^2 \leq b$ we have for any $a > 0$

$$\begin{aligned}
\|x_{k+1} - x_*\|^2 &= \|x_{k+1} - \tilde{x}_{k+1} + \tilde{x}_{k+1} - x_*\|^2 \\
&\leq \left(1 + a^{-1}\right)\|x_{k+1} - \tilde{x}_{k+1}\|^2 + (1 + a)\|\tilde{x}_{k+1} - x_*\|^2 \\
&\leq \left(1 + a^{-1}\right)b + (1 + a)\|\tilde{x}_{k+1} - x_*\|^2.
\end{aligned}$$

Plugging in $a = \frac{\eta^2\mu^2}{1+2\eta\mu}$ we get

$$\|x_{k+1} - x_*\|^2 \leq \left(\frac{1 + \eta\mu}{\eta\mu}\right)^2 b + \frac{(1 + \eta\mu)^2}{1 + 2\eta\mu}\|\tilde{x}_{k+1} - x_*\|^2. \qquad (23)$$

For the second term in eq. (23), we have by Fact 1 that $x_* = \text{prox}_{\eta f_{m_k}}(x_* + \eta \nabla f_{m_k}(x_*))$, then using Fact 2 we get

$$\|\tilde{x}_{k+1} - x_*\|^2 = \left\| \text{prox}_{\eta f_{m_k}}(x_k - \eta g_k) - \text{prox}_{\eta f_{m_k}}(x_* + \eta \nabla f_{m_k}(x_*)) \right\|^2$$

$$\leq \frac{1}{(1+\eta\mu)^2} \|x_k - \eta g_k - (x_* + \eta \nabla f_{m_k}(x_*))\|^2.$$

Expanding out the square we have

$$\|\tilde{x}_{k+1} - x_*\|^2 \leq \frac{1}{(1+\eta\mu)^2} \|x_k - x_* - \eta (g_k + \nabla f_{m_k}(x_*))\|^2$$

$$= \frac{1}{(1+\eta\mu)^2} \left[ \|x_k - x_*\|^2 + \eta^2 \|g_k + \nabla f_{m_k}(x_*)\|^2 - 2\eta \langle x_k - x_*, g_k + \nabla f_{m_k}(x_*) \rangle \right].$$

We denote by $\mathbb{E}_k[\cdot]$ the expectation conditional on all information up to (and including) the iterate $x_k$, then

$$\mathbb{E}_k \left[ \|\tilde{x}_{k+1} - x_*\|^2 \right] \leq \frac{1}{(1+\eta\mu)^2} [\|x_k - x_*\|^2 + \eta^2 \mathbb{E}_k \left[ \|g_k + \nabla f_{m_k}(x_*)\|^2 \right] \tag{24}$$
$$- 2\eta \langle x_k - x_*, \mathbb{E}_k [g_k + \nabla f_{m_k}(x_*)] \rangle],$$

where in the last term the expectation went inside the inner product since the expectation is conditioned on knowledge of $x_k$, and the randomness in $m$ is independent of $x_k$. Note that this expectation can be computed as

$$\mathbb{E}_k [g_k + \nabla f_{m_k}(x_*)] = \mathbb{E}_k [\nabla f(w_k) - \nabla f_{m_k}(w_k) + \nabla f_{m_k}(x_*)]$$
$$= \nabla f(w_k) - \nabla f(w_k) + \nabla f(x_*)$$
$$= 0 + 0 = 0,$$

where we used that since $x_*$ minimizes $f$ we must have $\nabla f(x_*) = 0$. Plugging this into (24) gives

$$\mathbb{E}_k \left[ \|\tilde{x}_{k+1} - x_*\|^2 \right] \leq \frac{1}{(1+\eta\mu)^2} \left[ \|x_k - x_*\|^2 + \eta^2 \mathbb{E}_k \left[ \|g_k + \nabla f_{m_k}(x_*)\|^2 \right] \right]. \tag{25}$$

For the second term, we can add $\nabla f(x_*)$ term inside (as it is a zero) to get

$$\mathbb{E}_k \left[ \|g_k + \nabla f_{m_k}(x_*)\|^2 \right] = \mathbb{E}_k \left[ \|g_k + \nabla f_{m_k}(x_*) - \nabla f(x_*)\|^2 \right]$$

$$= \mathbb{E}_k \left[ \|\nabla f(w_k) - \nabla f_{m_k}(w_k) + \nabla f_{m_k}(x_*) - \nabla f(x_*)\|^2 \right]$$

$$= \mathbb{E}_k \left[ \|\nabla f(w_k) - \nabla f_m(w_k) - [\nabla f(x_*) - \nabla f_m(x_*)]\|^2 \right]$$

$$= \frac{1}{M} \sum_{m=1}^{M} \|\nabla f(w_k) - \nabla f_m(w_k) - [\nabla f(x_*) - \nabla f_m(x_*)]\|^2. \tag{26}$$

Using Assumption 1 with eq. (26) we have

$$\mathbb{E}_k \left[ \|g_k + \nabla f_m(x_*)\|^2 \right] \leq \frac{1}{M} \sum_{m=1}^{M} \|\nabla f(w_k) - \nabla f_m(w_k) - [\nabla f(x_*) - \nabla f_m(x_*)]\|^2$$

$$\leq \delta^2 \|w_k - x_*\|^2.$$

Hence we can bound (25) as

$$\mathbb{E}_k \left[ \|\tilde{x}_{k+1} - x_*\|^2 \right] \leq \frac{1}{(1+\eta\mu)^2} \left[ \|x_k - x_*\|^2 + \eta^2 \delta^2 \|w_k - x_*\|^2 \right]. \tag{27}$$

Taking conditional expectation in eq. (23) and plugging the estimate of eq. (27) in we get

$$\mathbb{E}_k \left[ \|x_{k+1} - x_*\|^2 \right] \leq \left( \frac{1+\eta\mu}{\eta\mu} \right)^2 b + \frac{(1+\eta\mu)^2}{1+2\eta\mu} \mathbb{E}_k \left[ \|\tilde{x}_{k+1} - x_*\|^2 \right]$$

$$\leq \left( \frac{1+\eta\mu}{\eta\mu} \right)^2 b + \frac{1}{1+2\eta\mu} \left[ \|x_k - x_*\|^2 + \eta^2 \delta^2 \|w_k - x_*\|^2 \right]. \tag{28}$$

Observe that by design we have

$$\mathbb{E}_k\left[\|w_{k+1} - x_*\|^2\right] = p \cdot \|x_{k+1} - x_*\|^2 + (1-p) \cdot \|w_k - x_*\|^2. \tag{29}$$

Let $\alpha = \frac{\eta\mu}{p}$, then using eqs. (28) and (29) we have

$$\mathbb{E}_k\left[\|x_{k+1} - x_*\|^2\right] + \alpha\mathbb{E}_k\left[\|w_{k+1} - x_*\|^2\right] = (1+\alpha p)\mathbb{E}_k\left[\|x_{k+1} - x_*\|^2\right] + \alpha(1-p) \cdot \|w_k - x_*\|^2$$

$$\leq \frac{1+\alpha p}{1+2\eta\mu}\left[\|x_k - x_*\|^2 + \eta^2\delta^2\|w_k - x_*\|^2\right] + \alpha(1-p) \cdot \|w_k - x_*\|^2 + (1+\alpha p)\left(\frac{1+\eta\mu}{\eta\mu}\right)^2 b$$

$$= \frac{1+\alpha p}{1+2\eta\mu}\left[\|x_k - x_*\|^2 + \eta^2\delta^2\|w_k - x_*\|^2\right] + \alpha(1-p) \cdot \|w_k - x_*\|^2 + \frac{(1+\eta\mu)^3}{(\eta\mu)^2}b$$

$$= \frac{1+\alpha p}{1+2\eta\mu}\|x_k - x_*\|^2 + \alpha\left(1 - p + \frac{\eta^2\delta^2(1+\alpha p)}{\alpha(1+2\eta\mu)}\right)\|w_k - x_*\|^2 + \frac{(1+\eta\mu)^3}{(\eta\mu)^2}b$$

$$= \frac{1+\eta\mu}{1+2\eta\mu}\|x_k - x_*\|^2 + \alpha\left(1 - p + \frac{p\eta\delta^2}{\mu}\frac{1+\eta\mu}{1+2\eta\mu}\right)\|w_k - x_*\|^2 + \frac{(1+\eta\mu)^3}{(\eta\mu)^2}b. \tag{30}$$

Note that by condition on the stepsize we have $\eta\delta^2/\mu \leq \frac{1}{2}$, hence

$$\frac{\eta\delta^2}{\mu} \cdot \frac{1+\eta\mu}{1+2\eta\mu} \leq \frac{1}{2}\frac{1+\eta\mu}{1+2\eta\mu} \leq \frac{1}{2} \cdot 1 = \frac{1}{2}.$$

Using this in the second term of eq. (30) gives

$$\mathbb{E}_k\left[\|x_{k+1} - x_*\|^2\right] + \alpha\mathbb{E}_k\left[\|w_{k+1} - x_*\|^2\right]$$

$$\leq \frac{1+\eta\mu}{1+2\eta\mu}\|x_k - x_*\|^2 + \alpha\left(1 - p + \frac{p}{2}\right)\|w_k - x_*\|^2 + \frac{(1+\eta\mu)^3}{(\eta\mu)^2}b$$

$$= \frac{1+\eta\mu}{1+2\eta\mu}\|x_k - x_*\|^2 + \alpha\left(1 - \frac{p}{2}\right)\|w_k - x_*\|^2 + \frac{(1+\eta\mu)^3}{(\eta\mu)^2}b$$

$$\leq \max\left\{\frac{1+\eta\mu}{1+2\eta\mu}, 1 - \frac{p}{2}\right\}\left[\|x_k - x_*\|^2 + \alpha\|w_k - x_*\|^2\right] + \frac{(1+\eta\mu)^3}{(\eta\mu)^2}b.$$

Define the Lyapunov function $V_k = \|x_k - x_*\|^2 + \frac{\eta\mu}{p}\|w_k - x_*\|^2$. Then the last equation can simply be written as

$$\mathbb{E}_k\left[V_{k+1}\right] \leq \max\left\{\frac{1+\eta\mu}{1+2\eta\mu}, 1 - \frac{p}{2}\right\} \cdot V_k + \frac{(1+\eta\mu)^3}{(\eta\mu)^2}b.$$

Taking unconditional expectation gives

$$\mathbb{E}\left[V_{k+1}\right] \leq \max\left\{\frac{1+\eta\mu}{1+2\eta\mu}, 1 - \frac{p}{2}\right\}\mathbb{E}\left[V_k\right] + \frac{(1+\eta\mu)^3}{(\eta\mu)^2}b.$$

Let $\tau = \min\{\frac{\eta\mu}{1+2\eta\mu}, \frac{p}{2}\}$, then $\max\left\{\frac{1+\eta\mu}{1+2\eta\mu}, 1 - \frac{p}{2}\right\} = 1 - \tau$, and we get the simple recursion

$$\mathbb{E}\left[V_{k+1}\right] \leq (1-\tau)\mathbb{E}\left[V_k\right] + \frac{(1+\eta\mu)^3}{(\eta\mu)^2}b.$$

Iterating this for $k$ steps and using the formula for the sum of the geometric series gives for any $k \leq K$,

$$\mathbb{E}\left[V_k\right] \leq (1-\tau)^k\mathbb{E}\left[V_0\right] + \frac{(1+\eta\mu)^3}{(\eta\mu)^2}b\sum_{t=0}^{k-1}(1-\tau)^t$$

$$\leq (1-\tau)^k\mathbb{E}\left[V_0\right] + \frac{(1+\eta\mu)^3}{(\eta\mu)^2}b\sum_{t=0}^{\infty}(1-\tau)^t$$

$$= (1-\tau)^k\mathbb{E}\left[V_0\right] + \frac{(1+\eta\mu)^3}{(\eta\mu)^2\tau}b. \tag{31}$$

Now note that

$$\mathbb{E}\left[\|x_k - x_*\|^2\right] \leq \mathbb{E}\left[V_k\right]. \tag{32}$$

And by initialization we have $w_0 = x_0$, hence

$$\mathbb{E}\left[V_0\right] = \|x_0 - x_*\|^2 + \frac{\eta\mu}{p}\|w_0 - x_*\|^2 = \left(1 + \frac{\eta\mu}{p}\right)\|x_0 - x_*\|^2. \tag{33}$$

Plugging eqs. (32) and (33) into eq. (31) gives for any $k \leq K$,

$$\mathbb{E}\left[\|x_k - x_*\|^2\right] \leq \left(1 + \frac{\eta\mu}{p}\right)(1 - \tau)^k\|x_0 - x_*\|^2 + \frac{(1 + \eta\mu)^3}{(\eta\mu)^2\tau}b. \tag{34}$$

For the second statement of the theorem, observe that by assumption on $b$ we can bound the right hand side of eq. (34) as

$$\mathbb{E}\left[\|x_k - x_*\|^2\right] \leq \left(1 + \frac{\eta\mu}{p}\right)(1 - \tau)^k\|x_0 - x_*\|^2 + \frac{\epsilon}{2}.$$

Next, we use that for all $x \geq 0$ we have $1 - x \leq \exp(-x)$, hence we have after $K$ iterations of SVRP that

$$\mathbb{E}\left[\|x_K - x_*\|^2\right] \leq \left(1 + \frac{\eta\mu}{p}\right)\exp(-\tau K)\|x_0 - x_*\|^2 + \frac{\epsilon}{2}.$$

Thus if we run for the following number of iterations

$$K \geq \frac{1}{\tau}\log\left(\frac{2\|x_0 - x_*\|^2\left(1 + \frac{\eta\mu}{p}\right)}{\epsilon}\right) \tag{35}$$

We get that $\mathbb{E}\left[\|x_K - x_*\|^2\right] \leq \frac{\epsilon}{2} + \frac{\epsilon}{2} = \epsilon$. Choose $\eta = \frac{\mu}{2\delta^2}$ and $p = \frac{1}{M}$, then eq. (35) reduces to

$$K \geq 2\max\left\{\frac{\delta^2}{\mu^2} + 1, M\right\}\log\left(\frac{2\|x_0 - x_*\|^2\left(1 + \frac{\mu^2 M}{2\delta^2}\right)}{\epsilon}\right).$$

This gives the second part of the theorem. ∎

## G  PROOFS FOR CATALYST+SVRP

The convergence rate of Catalyst is given by the following proposition:

**Proposition 1.** *(Catalyst convergence rate). Run Catalyst (Algorithm 3) with smoothing parameter $\gamma$ for a $\mu$-strongly convex function $f$. Let $q = \frac{\mu}{\mu+\gamma}$ and choose $\rho \leq \sqrt{q}$. Assume at each timestep $t = 1, 2, \ldots, T$ we have that $x_t$ from eq. (36) satisfies*

$$\mathbb{E}\left[h_t(x_t) - \min_{x \in \mathbb{R}^d} h_t(x)\right] \leq \epsilon_t \overset{def}{=} \frac{2}{9}\left(f(x_0) - f(x_*)\right)(1 - \rho)^t. \tag{37}$$

*Then the iterates generated by Algorithm 3 satisfy*

$$\mathbb{E}\left[f(x_t) - f(x_*)\right] \leq \frac{8}{(\sqrt{q} - \rho)^2}(1 - \rho)^{t+1}\left(f(x_0) - f(x_*)\right). \tag{38}$$

*Proof.* This is (Lin et al., 2015, Theorem 3.1). ∎

We see that in order to apply Catalyst, we have to solve the problem point iterations of eq. (36) up to the accuracy given by eq. (37). However, the accuracy $\epsilon_t$ depends on the suboptimality gap $f(x_0) - f(x_*)$, which we do not have access to in general. The next proposition shows that this is not a problem for methods with linear convergence, and we can instead run the method for a fixed number of iterations:

---

**Algorithm 3:** Catalyst

---

**Data:** Initialization $x_0$, smoothing parameter $\gamma$, algorithm $\mathcal{A}$, strong convexity constant $\mu$.

1   Initialize $y_0 = x_0$.

2   Initialize $q = \frac{\mu}{\mu+\gamma}$ and $\alpha_0 = \sqrt{q}$.

3   **for** $t = 1, 2, \ldots, T-1$ **do**

4       Find $x_t$ using $\mathcal{A}$ starting from the initialization $x_{t-1}$:

$$x_t \approx \underset{x \in \mathbb{R}^d}{\operatorname{argmin}} \left\{ h_t(x) \overset{\text{def}}{=} f(x) + \frac{\gamma}{2}\|x - y_{t-1}\|^2 \right\}. \tag{36}$$

5       Update $\alpha_t \in (0,1)$ as

$$\alpha_t^2 = (1 - \alpha_t)\alpha_{t-1}^2 + q\alpha_t.$$

6       Compute $y_t$ using an extrapolation step

$$y_t = x_t + \beta_t(x_t - x_{t-1}) \qquad \text{with } \beta_t = \frac{\alpha_{t-1}(1 - \alpha_{t-1})}{\alpha_{t-1}^2 + \alpha_t}.$$

---

**Proposition 2.** *(Inner method complexity in Catalyst). Consider Algorithm 3 run with smoothing parameter $\gamma$ on a $\mu$-strongly convex function $f$, with $q = \frac{\mu}{\mu+\gamma}$ and $\rho \leq \sqrt{q}$. Suppose that the method $\mathcal{A}$ generates iterates $(z_s)_{s \geq 0}$ such that*

$$\mathbb{E}\left[h_t(z_s) - \min_{x \in \mathbb{R}^d} h_t(x)\right] \leq A(1 - \tau_{\mathcal{A},h})^s \left(h_t(z_0) - \min_{x \in \mathbb{R}^d} h_t(x)\right), \tag{39}$$

*then the precision $\epsilon_t$ from eq. (37) is reached in expectation when the number of iterations $s$ of $\mathcal{A}$ exceeds $T_{\mathcal{A}}$ where*

$$T_{\mathcal{A}} = \frac{1}{\tau_{\mathcal{A},h}} \log\left(A \cdot \left(\frac{2}{1-\rho} + \frac{2592\gamma}{\mu(1-\rho)^2(\sqrt{q}-\rho)^2}\right)\right).$$

*Proof.* This is (Lin et al., 2015, Proposition 3.2). ∎

Thus applying Catalyst to accelerating SVRP reduces to verifying if the condition in (39) holds for SVRP. The next proposition shows it does.

**Proposition 3.** *Define $h_t$ as in eq. (36) with $\gamma + \mu \leq \delta$, and let the objective $f : \mathbb{R}^d \to \mathbb{R}$ be a finite sum (as in eq. (1)) where each $f_m$ is $\mu$-strongly convex, $f$ is $L$-smooth, and Assumption 1 holds. Use SVRP (Algorithm 2) as the solver $\mathcal{A}$ to minimize $h_t$, then the iterates $z_1, z_2, \ldots, z_s$ generated by SVRP with stepsize $\eta = \frac{\mu+\gamma}{2\delta^2}$, solution accuracy $b = 0$, and communication probability $p = \frac{1}{M}$ satisfy*

$$\mathbb{E}\left[h_t(z_s) - \min_{x \in \mathbb{R}^d} h_t(x)\right] \leq A(1 - \tau)^s \left(h_t(z_0) - \min_{x \in \mathbb{R}^d} h_t(x)\right),$$

*where*

$$A \overset{\text{def}}{=} \frac{L+\gamma}{\mu+\gamma}\left(1 + \frac{(\gamma+\mu)^2 M}{\delta^2}\right), \qquad \text{and,} \qquad \tau \overset{\text{def}}{=} \frac{1}{2}\min\left\{\frac{1}{\frac{\delta^2}{(\gamma+\mu)^2}+1}, \frac{1}{M}\right\}.$$

*Proof.* The function $h_t$ (from Algorithm 3) is defined in eq. (36) as

$$h_t(x) = f(x) + \frac{\gamma}{2}\|x - y_{t-1}\|^2.$$

By the finite-sum structure, we can write $h_t$ as

$$h_t(x) = \frac{1}{M} \sum_{m=1}^{M} \left[f_m(x) + \frac{\gamma}{2}\|x - y_{t-1}\|^2\right], \tag{40}$$

where each $f_m$ is $\mu$-strongly convex. For each $m \in [M]$, define $h_{t,m}(x) \stackrel{\text{def}}{=} f_m(x) + \frac{\gamma}{2}\|x - y_{t-1}\|^2$. Note that $h_{t,m}$ is $\mu + \gamma$-strongly convex. Moreover, by direct computation we have for any $x, y \in \mathbb{R}^d$ that

$$\nabla h_{t,m}(x) - \nabla h_t(x) = \nabla f_m(x) + \gamma(x - y_{t-1}) - (\nabla f(x) + \gamma(x - y_{t-1}))$$
$$= \nabla f_m(x) - \nabla f(x).$$

Thus using this combined with Assumption 1 we have

$$\frac{1}{M} \sum_{m=1}^{M} \|\nabla h_{t,m}(x) - \nabla h_t(x) - [\nabla h_{t,m}(y) - \nabla h_t(y)]\|^2$$
$$= \frac{1}{M} \sum_{m=1}^{M} \|\nabla f_m(x) - \nabla f(x) - [\nabla f_m(y) - \nabla f(y)]\|^2$$
$$\leq \delta^2 \|x - y\|^2.$$

It follows that problem eq. (40) satisfies the conditions of Theorem 2 on the convergence of SVRP, and thus specializing the theorem we get for the iterates $z_1, z_2, \ldots, z_s$ by SVRP with stepsize $\eta = \frac{\mu+\gamma}{2\delta^2}$, solution accuracy $b = 0$, and communication probability $p = \frac{1}{M}$ that

$$\mathbb{E}\left[\|z_s - z^*\|^2\right] \leq \left(1 + \frac{(\mu+\gamma)^2 M}{2\delta^2}\right)(1 - \tau)^s \|z_0 - z^*\|^2, \tag{41}$$

where $\tau = \frac{1}{2}\min\left\{\frac{1}{\frac{\delta^2}{(\mu+\gamma)^2}+1}, \frac{1}{M}\right\}$ and $z^* = \arg\min_{x \in \mathbb{R}^d} h_t(x)$. Note that because $f$ is $L$-smooth and $\mu$-strongly convex, we have that $h_t$ is $L + \gamma$-smooth and $\mu + \gamma$-strongly convex, hence for any $x \in \mathbb{R}^d$ we have

$$h_t(x) - h_t(z^*) \geq \frac{\mu+\gamma}{2}\|x - z^*\|^2 \tag{42}$$

$$h_t(x) - h_t(z^*) \leq \frac{L+\gamma}{2}\|x - z^*\|^2. \tag{43}$$

Using (42) with $x = z_0$ and (43) with $x = z_s$ and combining with (41) we obtain

$$h_t(z_s) - h_t(z^*) \leq \frac{L+\gamma}{2}\|z_s - z^*\|^2$$
$$\leq \frac{L+\gamma}{2}\left(1 + \frac{(\mu+\gamma)^2 M}{2\delta^2}\right)(1 - \tau)^s \|z_0 - z^*\|^2$$
$$\leq \frac{L+\gamma}{\mu+\gamma}\left(1 + \frac{(\mu+\gamma)^2 M}{2\delta^2}\right)(1 - \tau)^s (h_t(z_0) - h_t(z^*)),$$

where $\tau = \frac{1}{2}\min\left\{\frac{1}{\frac{\delta^2}{(\mu+\gamma)^2}+1}, \frac{1}{M}\right\}$ and $z^*$ minimizes $h_t$. ∎

### G.1 PROOF OF THEOREM 3

*Proof.* The proof of this theorem is a straighfrforward combination of Propositions 1, 2, and 3. We set $p = \frac{1}{M}$, $b = 0$, and shall set $\eta$ and $\gamma$ later. In Proposition 1, choose $\rho = \frac{\sqrt{q}}{2} = \frac{\sqrt{\mu/(\mu+\gamma)}}{2}$, then the convergence guarantee in eq. (38) is

$$\mathbb{E}[f(x_t) - f(x_*)] \leq \frac{32}{q}\left(1 - \frac{\sqrt{q}}{2}\right)^{t+1}(f(x_0) - f(x_*))$$
$$= \frac{32(\mu+\gamma)}{\mu}\left(1 - \frac{\sqrt{q}}{2}\right)^{t+1}(f(x_0) - f(x_*))$$
$$\leq \frac{32(\mu+\gamma)}{\mu}\exp\left(-\frac{\sqrt{q}}{2}(t+1)\right)(f(x_0) - f(x_*)),$$

where in the last line we used that $1 - x \leq \exp(-x)$. Thus in order to get an $\epsilon$-approximate solution the the number of iterations of Catalyst $\mathrm{T}_{\text{iter}}^{\text{Catalyst}}$ should be

$$
\begin{aligned}
\mathrm{T}_{\text{iter}}^{\text{Catalyst}} &= \frac{2}{\sqrt{q}} \log\left( \frac{f(x_0) - f(x_*)}{\epsilon} \frac{32(\mu + \gamma)}{\mu} \right) \\
&= 2\sqrt{\frac{\mu + \gamma}{\mu}} \log\left( \frac{f(x_0) - f(x_*)}{\epsilon} \frac{32(\mu + \gamma)}{\mu} \right).
\end{aligned}
$$

By Proposition 2, for a method $\mathcal{A}$ satisfying eq. (39) we need $T_{\mathcal{A}}$ inner loop iterations, where $T_{\mathcal{A}}$ is defined as

$$
T_{\mathcal{A}} = \frac{1}{\tau_{\mathcal{A},h}} \log\left( A \cdot \left( \frac{2}{1 - \rho} + \frac{2592\gamma}{\mu(1 - \rho)^2(\sqrt{q} - \rho)^2} \right) \right).
$$

where $\tau_{\mathcal{A},h}$ and $A$ are defined as in eq. (39). By Proposition 3 we have that for SVRP with $\gamma + \mu \leq \delta$ and stepsize $\eta = \frac{\mu + \gamma}{2\delta^2}$ and communication probability $p = \frac{1}{M}$ that eq. (39) holds with

$$
A = \frac{L + \gamma}{\mu + \gamma} \left( 1 + \frac{(\gamma + \mu)^2 M}{\delta^2} \right), \qquad \text{and,} \qquad \tau = \min\left\{ \frac{1}{2\frac{\delta^2}{(\gamma + \mu)^2} + 2}, \frac{1}{2M} \right\}.
$$

Thus the number of iterations of SVRP $T_{\mathcal{A}}$ is

$$
T_{\mathcal{A}} = \max\left\{ 2\frac{\delta^2}{(\gamma + \mu)^2} + 2, 2M \right\} \log\left( A \cdot \left( \frac{2}{1 - \rho} + \frac{2592\gamma}{\mu(1 - \rho)^2(\sqrt{q} - \rho)^2} \right) \right).
$$

Thus the total number of SVRP iterations is $T_{\mathcal{A}} \times \mathrm{T}_{\text{iter}}^{\text{Catalyst}}$ which is

$$
\begin{aligned}
T_{\text{iter}}^{\text{total}} = 2\sqrt{\frac{\mu + \gamma}{\mu}} \max\left\{ \frac{2\delta^2}{(\gamma + \mu)^2} + 2, 2M \right\} &\log\left( A \cdot \left( \frac{2}{1 - \rho} + \frac{2592\gamma}{\mu(1 - \rho)^2(\sqrt{q} - \rho)^2} \right) \right) \\
&\times \log\left( \frac{f(x_0) - f(x_*)}{\epsilon} \frac{32(\mu + \gamma)}{\mu} \right).
\end{aligned}
$$

Let $\iota$ collect all the log factors, and use the looser estimate

$$
\max\left\{ \frac{2\delta^2}{(\gamma + \mu)^2} + 2, 2M \right\} \leq 4\max\left\{ \frac{\delta^2}{(\gamma + \mu)^2}, M \right\}
$$

which holds because $\gamma + \mu \leq \delta$ in all cases. Thus we get an $\epsilon$-accurate solution if the total number of iterations is equal to or exceeds

$$
T_{\text{iter}}^{\text{total}} = 8\sqrt{\frac{\gamma + \mu}{\mu}} \max\left\{ \frac{\delta^2}{(\gamma + \mu)^2}, M \right\} \iota.
$$

We now have two cases: (a) if $\frac{\delta}{\mu} \geq \sqrt{M}$, then the choice $\gamma = \sqrt{\frac{\delta^2}{M}} - \mu$ gives

$$
T_{\text{iter}}^{\text{total}} = 8M^{3/4}\sqrt{\frac{\delta}{\mu}}\iota. \tag{44}
$$

Note that under this choice of $\gamma$ we have $\gamma + \mu = \frac{\delta}{\sqrt{M}} \leq \delta$, and hence the precondition of Proposition 3 holds. (b) Otherwise, choosing $\gamma = 0$ yields

$$
T_{\text{iter}}^{\text{total}} = 8M\iota. \tag{45}
$$

Here we have $\gamma + \mu = \mu \leq \delta$ by assumption, and hence the precondition of Proposition 3 holds, and our usage of it is justified. Thus we reach an $\epsilon$-accurate solution in both cases when the total number of iterations satisfies

$$
T_{\text{iter}}^{\text{total}} = 8\iota \max\left\{ M, M^{3/4}\sqrt{\frac{\delta}{\mu}} \right\}.
$$

Finally, it remains to notice that the expected number of communication steps (by the same reasoning as in Section 4.2) is $\mathbb{E}\left[ T_{\text{comm}}^{\text{total}} \right] = (2 + 3pM)T_{\text{iter}}^{\text{total}} = 5T_{\text{iter}}^{\text{total}}$. $\blacksquare$

## H    EXTENSION TO THE CONSTRAINED SETTING

We consider the constrained problem defined as follows: let $R$ be a convex constraint function with an easy-to-compute proximal operator (i.e. we can compute $\text{prox}_R(\cdot)$ easily), then the composite finite-sum minimization problem is

$$\min_{x \in \mathbb{R}^d} \left[ F(x) = f(x) + R(x) = \frac{1}{M} \sum_{m=1}^{M} f_m(x) + R(x) \right]. \tag{46}$$

The constrained optimization problem $\min_{x \in C} f(x)$ can be reduced to problem (46) by letting $R = \delta_C$ be the indicator function on the closed convex set $C$, and the proximal operator associated with $R$ in this case reduces to the projection on $C$ (Beck, 2017, Theorem 6.24).

We shall use the following variant of SVRP to solve this problem:

---
**Algorithm 4:** SVRP for composite optimization

**Data:** Stepsize $\eta$, initialization $x_0$, number of steps $K$, communication probability $p$, local solution accuracy $b$.

1  Initialize $w_0 = x_0$.
2  **for** $k = 0, 1, 2, \ldots, K - 1$ **do**
3      Sample $m_k$ uniformly at random from $[M]$.
4      Set

$$g_k = \nabla f(w_k) - \nabla f_{m_k}(w_k).$$

5      Compute a $b$-approximation of the stochastic proximal point operator associated with $f_{m_k}$:

$$x_{k+1} \simeq \text{prox}_{\eta f_{m_k} + \eta R} (x_k - \eta g_k). \tag{47}$$

6      Sample $c_k \sim \text{Bernoulli}(p)$ and update $w_{k+1} = \begin{cases} x_{k+1} & \text{if } c_k = 1, \\ w_k & \text{if } c_k = 0. \end{cases}$

---

The following theorem gives the convergence rate of Algorithm 4:

**Theorem 5.** *(Convergence of SVRP in the composite setting). Suppose that Assumptions 1 and that each $f_{m_k}$ is $\mu$-strongly convex, and let $x_*$ be the minimizer of Problem (46). Suppose that each $x_{k+1}$ is a $b$-approximation of the proximal (47). Let $\tau = \min \left\{ \frac{\eta\mu}{1+2\eta\mu}, \frac{p}{2} \right\}$. Set the parameters of Algorithm 4 as $\eta = \frac{\mu}{2\delta^2}$, $b \le \frac{\epsilon\tau(\eta\mu)^2}{2(1+\eta\mu)^3}$, and $p = \frac{1}{M}$. Then the final iterate $x_K$ satisfies $\mathbb{E}\left[ \|x_K - x_*\|^2 \right] \le \epsilon$ provided that the total number of iterations $K$ is larger than $T_{\text{iter}}$:*

$$T_{\text{iter}} = \tilde{\mathcal{O}} \left( \left( M + \frac{\delta^2}{\mu^2} \right) \log \frac{1}{\epsilon} \right).$$

We first discuss the computational and communication complexities incurred by Algorithm 4 and then give the proof of Theorem 5 afterwards.

**Computational Complexity**. Note that this algorithm requires evaluating the proximal operator eq. (47), this is equivalent to solving the local optimization problem

$$\min_{x \in \mathbb{R}^d} \left[ \left( f_{m_k}(x) + \frac{1}{2\eta} \|x - z_k\|^2 \right) + R(x) \right],$$

for $z_k = x_k - \eta g_k$. When each $f_{m_k}$ is $\mu$-strongly convex, this is a composite convex optimization problem where the smooth part is $L + \frac{1}{\eta}$-smooth and $\mu + \frac{1}{\eta}$-strongly convex, and where the constraint $r$ has an easy to compute proximal operator. This can be solved by accelerated proximal gradient descent (Schmidt et al., 2011, Proposition 4) to any desired accuracy $b$ in

$$\mathcal{O} \left( \sqrt{\frac{L + \frac{1}{\eta}}{\mu + \frac{1}{\eta}}} \log \frac{1}{b} \right)$$

gradient and $R$-proximal operator accesses.

**Communication complexity.** The communication cost of Algorithm 4 is exactly the same as that of ordinary SVRP, as the iteration complexity of the method remains exactly the same. Thus, the discussion in Section 4.2 applies here. The communication cost is therefore of order $\tilde{\mathcal{O}}\left(M + \frac{\delta^2}{\mu^2}\right)$ communications.

**Catalyzed SVRP.** Catalyst applies out-of-the-box to composite optimization (Lin et al., 2015), provided that we are able to solve the composite problems. As Theorem 5 shows, Algorithm 4 can do that. Therefore we can show that Catalyzed SVRP has the communication complexity $\tilde{\mathcal{O}}\left(M + M^{\frac{3}{4}}\sqrt{\frac{\delta}{\mu}}\right)$ in this setting as well. The proof for this rate is a straightforward extension of the proofs in Section G.

### H.1 PROOFS FOR THE COMPOSITE SETTING

**Fact 3.** *Let $h$ be a convex function (not necessarily differentiable) and $\eta > 0$. Let $x \in \mathbb{R}^d$, and suppose that $g \in \partial h(x)$ (that is, $g$ is a subgradient of $h$ at $x$), then we have*

$$\mathrm{prox}_{\eta h}(x + \eta g) = x$$

*Proof.* Solving the proximal is equivalent ot

$$\mathrm{prox}_{\eta h}(z) = \arg \min_{y \in \mathbb{R}^d} \left( h(y) + \frac{1}{2\eta}\|y - z\|^2 \right)$$

This is a strongly convex minimization problem for any $\eta > 0$, hence, by Fermat's optimality condition (Beck, 2017, Theorem 3.63) the (necessarily unique) minimizer of this problem satisfies the first-order optimality condition

$$0 \in \partial \left( h(y) + \frac{1}{2\eta}\|y - z\|^2 \right) \tag{48}$$

Note that for any two proper convex functions $f_1, f_2$ on $\mathbb{R}^d$ we have that the subdifferential set of their sum $\partial(f_1 + f_2)(x)$ is the sum of points in their respective subdifferential sets (Beck, 2017, Theorem 3.36), i.e.

$$\partial(f_1 + f_2)(x) = \{g_1 + g_2 \mid g_1 \in \partial f_1(x), g_2 \in \partial f_2(x)\} = \partial f_1(x) + \partial f_2(x).$$

Thus the optimality condition eq. (48) reduces to

$$0 \in \partial h(y) + \frac{1}{\eta}\left[y - z\right].$$

Now observe that plugging $y = x$ and $z = x + \eta g$ and using the fact that $g \in \partial h(x)$ we have

$$g + \frac{1}{\eta}\left(x - (x + \eta g)\right) = g + \frac{-\eta g}{\eta} = 0.$$

It follows that $0 \in \partial h(x) + \frac{1}{\eta}\left[x - (x + \eta g)\right]$ and hence $\mathrm{prox}_{\eta h}(x + \eta g) = x$. $\blacksquare$

**Fact 4.** *(Tight contractivity of the proximal operator). If $h$ is $\mu$-strongly convex, then for all $\eta > 0$ and for any $x, y \in \mathbb{R}^d$ we have*

$$\left\|\mathrm{prox}_{\eta h}(x) - \mathrm{prox}_{\eta h}(y)\right\|^2 \leq \frac{1}{(1 + \eta\mu)^2}\|x - y\|^2$$

*Proof.* This is the nonsmooth generalization of Fact 2. Note that $p(x) = \mathrm{prox}_{\eta h}(x)$ satisfies $\eta g_{px} + [p(x) - x] = 0$ for some $g_{px} \in \partial h(p(x))$, or equivalently $p(x) = x - \eta g_{px}$. Using this we have

$$\begin{aligned}
\|p(x) - p(y)\|^2 &= \|[x - \eta g_{px}] - [y - \eta g_{py}]\|^2 \\
&= \|[x - y] - \eta[g_{px} - g_{py}]\|^2 \\
&= \|x - y\|^2 + \eta^2\|g_{px} - g_{py}\|^2 - 2\eta\langle x - y, g_{px} - g_{py}\rangle. \tag{49}
\end{aligned}$$

Now note that

$$\langle x - y, g_{px} - g_{py} \rangle = \langle p(x) + \eta g_{px} - [p(y) + \eta g_{py}], g_{px} - g_{py} \rangle$$
$$= \langle p(x) - p(y), g_{px} - g_{py} \rangle + \eta \|g_{px} - g_{py}\|^2. \tag{50}$$

Combining eqs. (49) and (50) we get

$$\|p(x) - p(y)\|^2 = \|x - y\|^2 + \eta^2 \|g_{px} - g_{py}\|^2 - 2\eta \langle p(x) - p(y), g_{px} - g_{py} \rangle$$
$$- 2\eta^2 \|g_{px} - g_{py}\|^2$$
$$= \|x - y\|^2 - \eta^2 \|g_{px} - g_{py}\|^2 - 2\eta \langle p(x) - p(y), g_{px} - g_{py} \rangle. \tag{51}$$

Let $D_h(u, v, g_v) = h(u) - h(v) - \langle g_v, u - v \rangle$ be the Bregman divergence associated with $h$ at $u, v$ with subgradient $g_v \in \partial h(v)$. It is easy to show that

$$\langle u - v, g_u - g_v \rangle = D_h(u, v, g_u) + D_h(v, u, g_v).$$

Using this with $u = p(x)$, $v = p(y)$, $g_u = g_{px}$ and $g_v = g_{py}$ and plugging back into (51) we get

$$\|p(x) - p(y)\|^2 = \|x - y\|^2 - \eta^2 \|g_{px} - g_{py}\|^2 - 2\eta \left[ D_h(p(x), p(y), g_{px}) + D_h(p(y), p(x), g_{py}) \right].$$

Note that because $h$ is strongly convex, we have by (Beck, 2017, Theorem 5.24 (ii)) that $D_h(p(y), p(x), g_{py}) \geq \frac{\mu}{2} \|p(y) - p(x)\|^2$ and $D_h(p(x), p(y), g_{px}) \geq \frac{\mu}{2} \|p(y) - p(x)\|^2$, hence

$$\|p(x) - p(y)\|^2 \leq \|x - y\|^2 - \eta^2 \|g_{px} - g_{py}\|^2 - 2\eta\mu \|p(x) - p(y)\|^2. \tag{52}$$

Strong convexity implies for any two points $u, v$ and $g_u \in \partial h(u)$, $g_v \in \partial h(v)$ that

$$\langle g_u - g_v, u - v \rangle \geq \mu \|u - v\|^2$$

see (Beck, 2017, Theorem 5.24 (iii)) for a proof. Using Cauchy-Schwartz yields

$$\|g_u - g_v\| \|u - v\| \geq \langle g_u - g_v, u - v \rangle \geq \mu \|u - v\|^2$$

Now if $u = v$, then trivially we have $\|g_u - g_v\| \geq \mu \|u - v\|$, otherwise, we divide both sides of the last inequality by $\|u - v\|$ to get

$$\|g_u - g_v\| \geq \mu \|u - v\|.$$

Thus in both cases we get

$$\|g_u - g_v\|^2 \geq \mu^2 \|u - v\|^2.$$

Using this in eq. (52) with $u = p(x)$, $v = p(y)$, $g_u = g_{px}$ and $g_v = g_{py}$ yields

$$\|p(x) - p(y)\|^2 \leq \|x - y\|^2 - \eta^2\mu^2 \|p(x) - p(y)\|^2 - 2\eta\mu \|p(x) - p(y)\|^2.$$

Rearranging gives

$$\left[1 + \eta^2\mu^2 + 2\eta\mu\right] \|p(x) - p(y)\|^2 \leq \|x - y\|^2.$$

It remains to notice that $(1 + \eta\mu)^2 = 1 + \eta^2\mu^2 + 2\eta\mu$. ∎

***Proof of Theorem 5.*** Let $\tilde{x}_{k+1} = \operatorname{prox}_{\eta f_{m_k} + \eta R}(x_k - \eta g_k)$. Then by eq. (8) and our assumption that $\|x_{k+1} - \tilde{x}_{k+1}\|^2 \leq b$ we have for any $a > 0$

$$\|x_{k+1} - x_*\|^2 = \|x_{k+1} - \tilde{x}_{k+1} + \tilde{x}_{k+1} - x_*\|^2$$
$$\leq \left(1 + a^{-1}\right) \|x_{k+1} - \tilde{x}_{k+1}\|^2 + (1 + a) \|\tilde{x}_{k+1} - x_*\|^2$$
$$\leq \left(1 + a^{-1}\right) b + (1 + a) \|\tilde{x}_{k+1} - x_*\|^2.$$

Plugging in $a = \frac{\eta^2\mu^2}{1 + 2\eta\mu}$ we get

$$\|x_{k+1} - x_*\|^2 \leq \left(\frac{1 + \eta\mu}{\eta\mu}\right)^2 b + \frac{(1 + \eta\mu)^2}{1 + 2\eta\mu} \|\tilde{x}_{k+1} - x_*\|^2. \tag{53}$$

Now observe that by first-order optimality of $x_*$ we have

$$0 \in \partial F(x_*) = \nabla f(x_*) + \partial R(x_*)$$

It follows that $-\nabla f(x_*) \in \partial R(x_*)$, and thus we can apply Fact 3 to get that $x_* = \text{prox}_{\eta f_{m_k} + \eta R}(x_* + \eta \nabla f_{m_k}(x_*) - \eta \nabla f(x_*))$. Using this in the second term in eq. (53) followed by Fact 4 we get

$$\|\tilde{x}_{k+1} - x_*\|^2 = \left\| \text{prox}_{\eta f_{m_k}}(x_k - \eta g_k) - \text{prox}_{\eta f_{m_k}}(x_* + \eta \nabla f_{m_k}(x_*) - \eta \nabla f(x_*)) \right\|^2$$

$$\leq \frac{1}{(1 + \eta\mu)^2} \|x_k - \eta g_k - (x_* + \eta \nabla f_{m_k}(x_*) - \eta \nabla f(x_*))\|^2.$$

Expanding out the square we have

$$\|\tilde{x}_{k+1} - x_*\|^2 \leq \frac{1}{(1 + \eta\mu)^2} \|x_k - x_* - \eta (g_k + \nabla f_{m_k}(x_*) - \nabla f(x_*))\|^2$$

$$= \frac{1}{(1 + \eta\mu)^2} \left[ \|x_k - x_*\|^2 + \eta^2 \|g_k + \nabla f_{m_k}(x_*) - \nabla f(x_*)\|^2 \right.$$

$$\left. - 2\eta \langle x_k - x_*, g_k + \nabla f_{m_k}(x_*) - \nabla f(x_*) \rangle \right].$$

We denote by $\mathbb{E}_k[\cdot]$ the expectation conditional on all information up to (and including) the iterate $x_k$, then

$$\mathbb{E}_k \left[ \|\tilde{x}_{k+1} - x_*\|^2 \right] \leq \frac{1}{(1 + \eta\mu)^2} [\|x_k - x_*\|^2 + \eta^2 \mathbb{E}_k \left[ \|g_k + \nabla f_{m_k}(x_*) - \nabla f(x_*)\|^2 \right]$$

$$- 2\eta \langle x_k - x_*, \mathbb{E}_k [g_k + \nabla f_{m_k}(x_*) - \nabla f(x_*)] \rangle], \tag{54}$$

where in the last term the expectation went inside the inner product since the expectation is conditioned on knowledge of $x_k$, and the randomness in $m$ is independent of $x_k$. Note that this expectation can be computed as

$$\mathbb{E}_k [g_k + \nabla f_{m_k}(x_*) - \nabla f(x_*)] = \mathbb{E}_k [\nabla f(w_k) - \nabla f_{m_k}(w_k) + \nabla f_{m_k}(x_*) - \nabla f(x_*)]$$

$$= \nabla f(w_k) - \nabla f(w_k) + \nabla f(x_*) - \nabla f(x_*)$$

$$= 0 + 0 = 0.$$

Plugging this into (54) gives

$$\mathbb{E}_k \left[ \|\tilde{x}_{k+1} - x_*\|^2 \right] \leq \frac{1}{(1 + \eta\mu)^2} \left[ \|x_k - x_*\|^2 + \eta^2 \mathbb{E}_k \left[ \|g_k + \nabla f_{m_k}(x_*) - \nabla f(x_*)\|^2 \right] \right]. \tag{55}$$

For the second term, we have

$$\mathbb{E}_k \left[ \|g_k + \nabla f_{m_k}(x_*) - \nabla f(x_*)\|^2 \right] = \mathbb{E}_k \left[ \|\nabla f(w_k) - \nabla f_{m_k}(w_k) + \nabla f_{m_k}(x_*) - \nabla f(x_*)\|^2 \right]$$

$$= \mathbb{E}_k \left[ \|\nabla f(w_k) - \nabla f_m(w_k) - [\nabla f(x_*) - \nabla f_m(x_*)]\|^2 \right]$$

$$= \frac{1}{M} \sum_{m=1}^{M} \|\nabla f(w_k) - \nabla f_m(w_k) - [\nabla f(x_*) - \nabla f_m(x_*)]\|^2. \tag{56}$$

Using Assumption 1 with eq. (56) we have

$$\mathbb{E}_k \left[ \|g_k + \nabla f_m(x_*)\|^2 \right] \leq \frac{1}{M} \sum_{m=1}^{M} \|\nabla f(w_k) - \nabla f_m(w_k) - [\nabla f(x_*) - \nabla f_m(x_*)]\|^2$$

$$\leq \delta^2 \|w_k - x_*\|^2.$$

Hence we can bound (55) as

$$\mathbb{E}_k \left[ \|\tilde{x}_{k+1} - x_*\|^2 \right] \leq \frac{1}{(1 + \eta\mu)^2} \left[ \|x_k - x_*\|^2 + \eta^2 \delta^2 \|w_k - x_*\|^2 \right]. \tag{57}$$

Taking conditional expectation in eq. (53) and plugging the estimate of eq. (57) in we get

$$\mathbb{E}_k\left[\|x_{k+1} - x_*\|^2\right] \leq \left(\frac{1+\eta\mu}{\eta\mu}\right)^2 b + \frac{(1+\eta\mu)^2}{1+2\eta\mu}\mathbb{E}_k\left[\|\tilde{x}_{k+1} - x_*\|^2\right]$$

$$\leq \left(\frac{1+\eta\mu}{\eta\mu}\right)^2 b + \frac{1}{1+2\eta\mu}\left[\|x_k - x_*\|^2 + \eta^2\delta^2\|w_k - x_*\|^2\right]. \quad (58)$$

Observe that by design we have

$$\mathbb{E}_k\left[\|w_{k+1} - x_*\|^2\right] = p\cdot\|x_{k+1} - x_*\|^2 + (1-p)\cdot\|w_k - x_*\|^2. \quad (59)$$

Let $\alpha = \frac{\eta\mu}{p}$, then using eqs. (58) and (59) we have

$$\mathbb{E}_k\left[\|x_{k+1} - x_*\|^2\right] + \alpha\mathbb{E}_k\left[\|w_{k+1} - x_*\|^2\right] = (1+\alpha p)\mathbb{E}_k\left[\|x_{k+1} - x_*\|^2\right] + \alpha(1-p)\cdot\|w_k - x_*\|^2$$

$$\leq \frac{1+\alpha p}{1+2\eta\mu}\left[\|x_k - x_*\|^2 + \eta^2\delta^2\|w_k - x_*\|^2\right] + \alpha(1-p)\cdot\|w_k - x_*\|^2 + (1+\alpha p)\left(\frac{1+\eta\mu}{\eta\mu}\right)^2 b$$

$$= \frac{1+\alpha p}{1+2\eta\mu}\left[\|x_k - x_*\|^2 + \eta^2\delta^2\|w_k - x_*\|^2\right] + \alpha(1-p)\cdot\|w_k - x_*\|^2 + \frac{(1+\eta\mu)^3}{(\eta\mu)^2}b$$

$$= \frac{1+\alpha p}{1+2\eta\mu}\|x_k - x_*\|^2 + \alpha\left(1 - p + \frac{\eta^2\delta^2(1+\alpha p)}{\alpha(1+2\eta\mu)}\right)\|w_k - x_*\|^2 + \frac{(1+\eta\mu)^3}{(\eta\mu)^2}b$$

$$= \frac{1+\eta\mu}{1+2\eta\mu}\|x_k - x_*\|^2 + \alpha\left(1 - p + \frac{p\eta\delta^2}{\mu}\frac{1+\eta\mu}{1+2\eta\mu}\right)\|w_k - x_*\|^2 + \frac{(1+\eta\mu)^3}{(\eta\mu)^2}b. \quad (60)$$

Note that by condition on the stepsize we have $\eta\delta^2/\mu \leq \frac{1}{2}$, hence

$$\frac{\eta\delta^2}{\mu}\cdot\frac{1+\eta\mu}{1+2\eta\mu} \leq \frac{1}{2}\frac{1+\eta\mu}{1+2\eta\mu} \leq \frac{1}{2}\cdot 1 = \frac{1}{2}.$$

Using this in the second term of eq. (60) gives

$$\mathbb{E}_k\left[\|x_{k+1} - x_*\|^2\right] + \alpha\mathbb{E}_k\left[\|w_{k+1} - x_*\|^2\right]$$

$$\leq \frac{1+\eta\mu}{1+2\eta\mu}\|x_k - x_*\|^2 + \alpha\left(1 - p + \frac{p}{2}\right)\|w_k - x_*\|^2 + \frac{(1+\eta\mu)^3}{(\eta\mu)^2}b$$

$$= \frac{1+\eta\mu}{1+2\eta\mu}\|x_k - x_*\|^2 + \alpha\left(1 - \frac{p}{2}\right)\|w_k - x_*\|^2 + \frac{(1+\eta\mu)^3}{(\eta\mu)^2}b$$

$$\leq \max\left\{\frac{1+\eta\mu}{1+2\eta\mu}, 1 - \frac{p}{2}\right\}\left[\|x_k - x_*\|^2 + \alpha\|w_k - x_*\|^2\right] + \frac{(1+\eta\mu)^3}{(\eta\mu)^2}b.$$

Define the Lyapunov function $V_k = \|x_k - x_*\|^2 + \frac{\eta\mu}{p}\|w_k - x_*\|^2$. Then the last equation can simply be written as

$$\mathbb{E}_k\left[V_{k+1}\right] \leq \max\left\{\frac{1+\eta\mu}{1+2\eta\mu}, 1 - \frac{p}{2}\right\}\cdot V_k + \frac{(1+\eta\mu)^3}{(\eta\mu)^2}b.$$

Taking unconditional expectation gives

$$\mathbb{E}\left[V_{k+1}\right] \leq \max\left\{\frac{1+\eta\mu}{1+2\eta\mu}, 1 - \frac{p}{2}\right\}\mathbb{E}\left[V_k\right] + \frac{(1+\eta\mu)^3}{(\eta\mu)^2}b.$$

Let $\tau = \min\{\frac{\eta\mu}{1+2\eta\mu}, \frac{p}{2}\}$, then $\max\left\{\frac{1+\eta\mu}{1+2\eta\mu}, 1 - \frac{p}{2}\right\} = 1 - \tau$, and we get the simple recursion

$$\mathbb{E}\left[V_{k+1}\right] \leq (1-\tau)\mathbb{E}\left[V_k\right] + \frac{(1+\eta\mu)^3}{(\eta\mu)^2}b.$$

Iterating this for $k$ steps and using the formula for the sum of the geometric series gives for any $k \leq K$,

$$
\begin{aligned}
\mathbb{E}\left[V_k\right] &\leq (1 - \tau)^k \mathbb{E}\left[V_0\right] + \frac{(1 + \eta\mu)^3}{(\eta\mu)^2} b \sum_{t=0}^{k-1} (1 - \tau)^t \\
&\leq (1 - \tau)^k \mathbb{E}\left[V_0\right] + \frac{(1 + \eta\mu)^3}{(\eta\mu)^2} b \sum_{t=0}^{\infty} (1 - \tau)^t \\
&= (1 - \tau)^k \mathbb{E}\left[V_0\right] + \frac{(1 + \eta\mu)^3}{(\eta\mu)^2\tau} b.
\end{aligned}
\tag{61}
$$

Now note that

$$
\mathbb{E}\left[\|x_k - x_*\|^2\right] \leq \mathbb{E}\left[V_k\right].
\tag{62}
$$

And by initialization we have $w_0 = x_0$, hence

$$
\mathbb{E}\left[V_0\right] = \|x_0 - x_*\|^2 + \frac{\eta\mu}{p}\|w_0 - x_*\|^2 = \left(1 + \frac{\eta\mu}{p}\right)\|x_0 - x_*\|^2.
\tag{63}
$$

Plugging eqs. (62) and (63) into eq. (61) gives for any $k \leq K$,

$$
\mathbb{E}\left[\|x_k - x_*\|^2\right] \leq \left(1 + \frac{\eta\mu}{p}\right)(1 - \tau)^k \|x_0 - x_*\|^2 + \frac{(1 + \eta\mu)^3}{(\eta\mu)^2\tau} b.
\tag{64}
$$

For the second statement of the theorem, observe that by assumption on $b$ we can bound the right hand side of eq. (64) as

$$
\mathbb{E}\left[\|x_k - x_*\|^2\right] \leq \left(1 + \frac{\eta\mu}{p}\right)(1 - \tau)^k \|x_0 - x_*\|^2 + \frac{\epsilon}{2}.
$$

Next, we use that for all $x \geq 0$ we have $1 - x \leq \exp(-x)$, hence we have after $K$ iterations of SVRP that

$$
\mathbb{E}\left[\|x_K - x_*\|^2\right] \leq \left(1 + \frac{\eta\mu}{p}\right)\exp(-\tau K)\|x_0 - x_*\|^2 + \frac{\epsilon}{2}.
$$

Thus if we run for the following number of iterations

$$
K \geq \frac{1}{\tau}\log\left(\frac{2\|x_0 - x_*\|^2\left(1 + \frac{\eta\mu}{p}\right)}{\epsilon}\right)
\tag{65}
$$

We get that $\mathbb{E}\left[\|x_K - x_*\|^2\right] \leq \frac{\epsilon}{2} + \frac{\epsilon}{2} = \epsilon$. Choose $\eta = \frac{\mu}{2\delta^2}$ and $p = \frac{1}{M}$, then eq. (65) reduces to

$$
K \geq 2\max\left\{\frac{\delta^2}{\mu^2} + 1, M\right\}\log\left(\frac{2\|x_0 - x_*\|^2\left(1 + \frac{\mu^2 M}{2\delta^2}\right)}{\epsilon}\right).
$$

This gives the second part of the theorem. ∎

# I    CLIENT-SERVER FORMULATIONS OF THE ALGORITHMS

The algorithms we gave (Algorithm 1 and Algorithm 2) are written in notation that does not make clear the exact role of server and client in the process. Below, we rewrite the algorithms to make clear the role of the clients and the server, in Algorithm 5 and Algorithm 2. Note that both formulations are exactly equivalent, and this is just for clarity. Both algorithms rely on evaluating the proximal operator approximately using local data: we give an example solution method using gradient descent in Algorithm 7. By standard convergence results for gradient descent, Algorithm 7 halts in at most $\mathcal{O}\left(\frac{L+\frac{1}{\eta}}{\mu+\frac{1}{\eta}}\log\frac{1}{\epsilon}\right)$ iterations. As discussed in the main text, we can also use accelerated gradient descent or other algorithms, this example is just for illustration.

---

**Algorithm 5:** Stochastic Proximal Point Method (SPPM) (Client-Server formulation)

---

**Data:** Stepsize $\eta$, initialization $x_0$, number of steps $K$, proximal solution accuracy $b$.

1 **Server** communicates stepsize $\eta$ and desired proximal solution accuracy $b$ to all clients.

2 **for** $k = 0, 1, 2, \ldots, K - 1$ **do**

3      **Server** samples client $m \in [M]$.

4      **Server** sends model $x_k$ to client $m$.

5      **Client** $m$ evaluates the local proximal operator up to accuracy $b$ on its local data (e.g. using Algorithm 7):
$$x_{k+1} \simeq \text{prox}_{\eta f_{m_k}}(x_k).$$

6      **Client** $m$ sends the updated model $x_{k+1}$ to the server.

---

---

**Algorithm 6:** Stochastic Variance-Reduced Proximal Point (SVRP) Method (Client-Server formulation)

---

**Data:** Stepsize $\eta$, initialization $x_0$, number of steps $K$, communication probability $p$, local solution accuracy $b$.

1 Initialize $w_0 = x_0$.

2 **Server** communicates stepsize $\eta$ and desired proximal solution accuracy $b$ to all clients.

3 **Server** communicates iterate $w_0$ to all clients.

4 **Every client** $m$ computes its local gradient $\nabla f_m(w_0)$ and sends it back to the server.

5 **Server** receives the local gradients and averages them: $\nabla f(w_0) = \frac{1}{M} \sum_{m=1}^{M} \nabla f_m(w_0)$.

6 **Server** sends $\nabla f(w_0)$ to all clients.

7 **for** $k = 0, 1, 2, \ldots, K - 1$ **do**

8      **Server** samples client $m_k$ uniformly at random from $[M]$.

9      **Server** sends model $x_k$ to client $m_k$.

10      **Client** $m_k$ computes its local gradient $\nabla f_{m_k}(w_k)$ with the (cached) model $w_k$ and cached full gradient $\nabla f(w_k)$ and sets

$$g_k = \nabla f(w_k) - \nabla f_{m_k}(w_k).$$

11      **Client** $m_k$ computes a $b$-approximation of the proximal point operator on its local data (e.g. using Algorithm 7):

$$x_{k+1} \simeq \mathrm{prox}_{\eta f_{m_k}} \left( x_k - \eta g_k \right).$$

12      **Client** $m_k$ sends $x_{k+1}$ to the server.

13      **Server** samples $c_k \sim \mathrm{Bernoulli}(p)$.

14      **if** $c_k = 1$ **then**

15          **Server** notifies clients of update, sets $w_{k+1} = x_{k+1}$ and sends it to all clients.

16          **Every client** $m$ caches the new model $w_{k+1}$ instead of the old model $w_k$, then computes its local gradient $\nabla f_m(w_{k+1})$ and sends it back to the server.

17          **Server** receives the local gradients and averages them:

         $\nabla f(w_{k+1}) = \frac{1}{M} \sum_{m=1}^{M} \nabla f_m(w_{k+1})$.

18          **Server** sends $\nabla f(w_{k+1})$ to all clients to cache.

19      **else**

20          **Clients** keep their cached vector $w_k$ as it is, setting $w_{k+1} = w_k$ automatically.

---

---

**Algorithm 7:** Client $m$ approximate proximal point evaluation using gradient descent.

---

**Data:** Proximal operator argument $z$, proximal solution accuracy $b$, smoothnes constant $L$, strong convexity constant $\mu$.

1   (*This algorithm evaluates* $\mathrm{prox}_{\eta f_m}(z)$ *up to accuracy $b$ using gradient descent.*)

2   Set the local stepsize $\beta = \frac{1}{L + \frac{1}{\eta}}$.

3   Set the model $y_0 = 0$.

4   **for** $t = 0, 1, 2, \ldots$ **do**

5     Compute the gradient

6

$$\Delta_t = \nabla f_m(y_t) + \frac{1}{\eta}(y_t - z).$$

7     Update the model

8

$$y_{t+1} = y_t - \beta \Delta_t.$$

9     **if** *the gradient norm is small* $\|\Delta_t\|^2 \le b \left[ \mu + \frac{1}{\eta} \right]^2$ **then**

10        **Exit** and return $y_t$ as it is a $b$-approximate solution, since by strong convexity we have

$$\left\| y_t - \mathrm{prox}_{\eta f_m}(z) \right\|^2 \le \frac{\|\Delta_t\|^2}{(\mu + \frac{1}{\eta})^2} \le b.$$

---

