# OpenReview forum: "Faster federated optimization under second-order similarity"
_ICLR.cc/2023/Conference — ICLR 2023 poster_

### Official Review · Reviewer_sQ1q · 2022-10-23

**Confidence:** 4
**Correctness:** 3
**Technical Novelty And Significance:** 3
**Empirical Novelty And Significance:** Not applicable
**Recommendation:** 5

**Clarity, Quality, Novelty And Reproducibility:**

This paper is well written.  However, the experiments are too toy to demonstrate the efficacy of the proposed approach.

**Strength And Weaknesses:**

The authors propose a new federated learning algorithm under the second-order similarity condition. Solid theoretical findings show the superiority of the proposed algorithm. However, before this work is accepted, several concerns should be resolved:
(i)	More discussion should be given on the second-order similarity. There exist several similarity conditions in analyzing the federated learning algorithms. Can the authors provide their relationships of the existing similarity conditions? In addition, do the results derived in this work still hold when assumption 1 is replaced by other similarity conditions? If not, can the authors provide several explanations?
(ii)	Existing experiments are too simple to demonstrate the practical effectiveness of the proposed approach. More comparisons with SOTA baselines such as FedSAM, FedDyn, FedCM, FedPD, and FedDR, should be included.
(iii)	In addition, the efficacy of SPPM and Catalyzed SVRP should be verified.



**Summary Of The Paper:**

In this work, the authors propose a new federated learning optimization based on variance reduced proximal point algorithm. Furthermore, a theoretical superiority of the proposed algorithm is derived. Preliminary experiments also demonstrate the effectiveness of the proposed algorithm.

**Summary Of The Review:**

see the comments above.

---

### Official Review · Reviewer_gy4c · 2022-10-23

**Confidence:** 2
**Clarity, Quality, Novelty And Reproducibility:** see above.
**Correctness:** 4
**Technical Novelty And Significance:** 3
**Empirical Novelty And Significance:** 1
**Recommendation:** 5

**Strength And Weaknesses:**

This paper is in general easy to read and the improvement seems significant under the corresponding assumptions. However, I have the following concerns:

1.	I am slightly confused about the setting: in federated learning, one important thing is protecting the privacy of the local learners, and this is the reason why the local learner only submits their model instead of their data. However, in this paper, if my understanding if correct, the sever need to compute the full gradient. Thus, does this mean all local clints have to submit their data? If the sever has access to all gradients, and the computational complexity of gradient computation is low, we do we still need the local clients? I think I may have some misunderstandings on this paper, and I hope the authors can help me understand this point.

2.	The experiments are not sufficient (for a stochastic optimization/federated learning paper). It is only conducted on one very simple data set. More experiments should be conducted to demonstrate the effectiveness of the proposed algorithms.

3.	The section on Catalyst is not well-developed: It seems that the Catalyst algorithm is just being plugged-in and it is hard to tell where the novelty is. Moreover, can the authors be clearer about how Algorithm 3 works in a sever-client setting?

4.	For assumption 1, I do not see why this inequality holds for the settings the authors mentioned. Can the authors elaborate on 1 or 2 examples?


A small question: In experiments, how do we compute x^*?

Some minor issues:
Page 2: ” several methods exist for methods exist for solving Problem”


**Summary Of The Paper:**

This paper studies Federated stochastic optimization with client sampling. Under the assumption of second-order similarity condition, the authors propose a variance-reduction based proximal point algorithm which enjoy a better convergence rate wrt the number of clients. They also proposed an accelerated version which enjoys an even tighter rate wrt to the condition number.



**Summary Of The Review:**

This paper is in general easy to read, and the improvement seems significant under the corresponding assumptions. However, I have some questions/concerns, which are listed above.

---

### Official Review · Reviewer_2cGY · 2022-10-24

**Confidence:** 4
**Correctness:** 4
**Technical Novelty And Significance:** 3
**Empirical Novelty And Significance:** 2
**Recommendation:** 6

**Clarity, Quality, Novelty And Reproducibility:**

The paper is very clear and easy to follow. The proofs are provided in great detail and correct to my understanding. The algorithms and analyses can provide novel insights into the community.

**Strength And Weaknesses:**

**Strength:**

The paper achieves better communication cost than related works for strongly-convex finite-sum federated optimization under second-order similarity, showing the benefit of client sampling and the trade-offs between local computation and global communication. The novel method that incorporates SVRG into the proximal point framework can also provide new algorithmic insights. Although the use of proximal point algorithms in federated optimization and the ability of catalyst framework to accelerate convex optimization is well-known in the literature, it is still interesting to see the improved rates in this work combining all the techniques.

**Weakness:**

1. It is not immediately clear whether the analyses can be extended to other settings, e.g. the constrained case because it is explicitly used that $\nabla f(x^*)=0$ in all the proofs. However, the original PPM (and also SGD) works for strongly-convex, convex, and nonconvex settings in both constrained and unconstrained regimes. The proposed algorithms also require extra memory for $g_k$.

2. As mentioned in the paper, the current catalyzed SVRP requires the exact proximal oracle, and it is not clear how to satisfy eq. (41) with the inexact proximal update. The double-looped (actually tripple-loop if considering local steps) catalyzed SVRP can be complicated to implement in practice. Will it be possible to directly accelerate SVRP to design a simpler algorithm?

3. It will be good to include the server and clients in the algorithms for a better understanding since some steps are only for clients and some are for the server. For the communication complexity, it should also take into account that the server needs to first communicate the stepsize $\eta$ and accuracy $b$ to all the clients even though it does not affect the final complexity.

4. More details about the experiments can be provided in the appendix, e.g. values for the stepsizes. What is the behavior if one tunes the stepsize for each algorithm instead of setting it to the optimal theoretical suggestion?

5. No experiments of catalyzed SVRP are provided. In the case that $\delta/\mu \leq\sqrt{M}$, SVRP is enough and the catalyst will not bring any benefit (is there a typo in the comment after Theorem 3 stating that Catalyzed SVRP is strictly better than SVRP when $\delta/\mu\leq\sqrt{M}$), and all experiments fall into this regime. It might be good to verify the behavior of catalyzed SVRP as well when $\delta$ is large.

6. Minor: The emphasis that SPPM does not require smoothness in the abstract can be misleading and not precise. Smoothness is required to solve the proximal operator efficiently when its closed form is unknown; It is better to include the example of differentially private optimization in the main text because it is also emphasized in the abstract; It is mentioned in the related work that Kovalev et al. (2022) remove logarithmic factors but it is still $\tilde O$ in the table; The symbol $\mathcal{M}$ in the catalyst might cause some confusion with $M$ for the number of clients; What is the relationship between second-order similarity and other smoothness conditions used in variance reduction, e.g. average smoothness and individual smoothness, and what are the benefits of introducing it?

7. Typos: "several methods exist for solving Problem (1)"; "SVRP trades off a higher lower computational complexity"; "Ryu & Boyd (2014) convergence rates"; $\eta=\mu\epsilon/(2\sigma^2)$? after Algorithm 1; Theorem 2: "and and that each $x_{k+1}$"; "We linear regression"; duplication in eq (17); Proposition 3 and proof of Theorem 3: should be $1/(\cdots+2)$ in $\tau$.

**Summary Of The Paper:**

The paper studies federated optimization under strong convexity and second-order similarity assumptions. Using the idea of trading off local computational cost for less communication, the paper proposes a new algorithm SVRP based on proximal point methods and variance reduction (SVRG). With client sampling, SVRP can achieve better communication cost compared to the state-of-the-art when the second-order similarity constant is small enough. Moreover, the catalyst acceleration of SVRP is also studied when assuming the exact proximal oracle. It attains strictly better complexity than all existing works.

**Summary Of The Review:**

The paper provides strictly better complexity for strongly-convex federated optimization under second-order similarity, with a novel algorithm SVRP and its catalyst acceleration. I am happy to increase the score based on the discussions.

---

### Official Review · Reviewer_Q8H3 · 2022-10-25

**Confidence:** 4
**Correctness:** 3
**Technical Novelty And Significance:** 2
**Empirical Novelty And Significance:** 2
**Recommendation:** 5

**Clarity, Quality, Novelty And Reproducibility:**

The writing can significantly be improved.
The novelty of this paper is limited. This paper basically combines
the existing approximate stochastic proximal point evaluations, client sampling, and variance reduction.

**Strength And Weaknesses:**

Strength:

This paper proposed two new SVRP and Catalyzed SVRP algorithms based on approximate stochastic proximal point evaluations, client sampling, and variance reduction. It provided the convergence analysis of the proposed algorithms, and
prove that the proposed Catalyzed-SVRP algorithm has a lower communication complexity  than the existing methods.

Weakness:

The novelty of this paper is limited. This paper basically combines the existing approximate stochastic proximal point evaluations, client sampling, and variance reduction.

**Summary Of The Paper:**

This paper studied the finite-sum federated optimization under a second-order function similarity condition and strong convexity, and proposed two new SVRP and Catalyzed SVRP algorithms based on approximate stochastic proximal point evaluations, client sampling, and variance reduction. It provided the convergence analysis of the proposed algorithms, and  prove that the proposed Catalyzed-SVRP algorithm has a lower communication complexity  than the existing methods. Some experimental results demonstrate the efficiency of the proposed algorithms.

**Summary Of The Review:**

This paper studied the finite-sum federated optimization under a second-order function similarity condition
and strong convexity, and proposed two new SVRP and Catalyzed SVRP algorithms
based on approximate stochastic proximal point evaluations, client sampling, and
variance reduction. It provided the convergence analysis of the proposed algorithms, and
prove that the proposed Catalyzed-SVRP algorithm has a lower communication complexity
than the existing methods. Some experimental results domenstrate the efficiency of the proposed algorithms.

Some Comments:

1. It would be great if the authors would detail the second-order function similarity condition.

2. It would be great if the authors would detail the reason of reducing the communication complexity in
the proposed methods.

3. There will be a strict upper limit of 9 pages for the main text of the submission,
with unlimited additional pages for citations (https://iclr.cc/Conferences/2023/CallForPapers).
This paper maybe not fit for this request.



--------------------------------------------------------------------------------------------------------------------------------------
--------------------------------------------------------------------------------------------------------------------------------------
The authors still did not solve my main concern- the limited novelty of this paper. So I keep my score.

---

### Official Review · Reviewer_JTga · 2022-10-26

**Confidence:** 3
**Clarity, Quality, Novelty And Reproducibility:** Overall, the paper is well-written wi…
**Correctness:** 3
**Technical Novelty And Significance:** 2
**Empirical Novelty And Significance:** 2
**Recommendation:** 5

**Strength And Weaknesses:**

* Strengths:

1.	Overall, the paper is well-written with a clear motivation.

2.	Using stochastic proximal point framework to unify the analysis is interesting.

3.	The authors provide an elegant combination of prior techniques used in the federated learning literature including client sampling, variance reduction, and stochastic proximal point method.

* Major concerns:

1.	(Computational cost) The analysis shows that under the partial participation setting, the proposed methods improve communication complexity. However, this gain is achieved at the expense of additional computation on the local clients. Even though the experiments demonstrate superior performance in terms of communication, no experiments are provided to compare the computation trade-off. It would be interesting to evaluate this trade-off empirically.

2.	(Small-scale experiments) Moreover, the experiments are provided on relatively simpler learning problems. It would be interesting to evaluate the methods on more complex federated learning problems, e.g. using logistic regression on image data, or training a neural network, as done by the authors of SCAFFOLD (Karimireddy et al., 2022)

3.	(Limited novelty) Since distributed optimization has already been studied under assumption 1 in the full participation setting, improving the existing algorithms under the partial participation setting seems rather limited in terms of novelty.

* Minor concerns:

1)	Typo: In the paragraph above section 1.1, the phrase “methods exist for” is redundant

2)	Missing word: In the third last paragraph of section 2, the sentence “Rye & Boyd (2014) convergence rates…” should instead be something like “Rye & Boyd (2014) provide convergence rates…”

3)	Figure 1: I found the font size to be very small. For better visibility, I would recommend turning the grid on and increasing the font size of the axis labels, and legends.

4)	Missing word: In the beginning of section 5, the sentence “We linear regression with…” should instead be something like “We run linear regression with…”.


**Summary Of The Paper:**

This paper studies the federated learning optimization problem under the assumption of second-order function similarity. In contrast to the previous work that considers the full client participation model, this work studies the problem under the partial participation/client sampling model. Two communication-efficient algorithms are proposed, namely SVRP and Catalyzed SVRP, which use the stochastic proximal point method as a building block and adapt it via an SVRG-style variance reduction method.

The authors provide analysis of the stochastic proximal point method without the smoothness assumption. Moreover, the analysis of the algorithms shows the improved communication complexity of the proposed methods as compared to the prior methods. Experiments on real and synthetic data show the improved performance of the proposed methods in terms of communication.

**Summary Of The Review:**

Overall, the idea of the paper is sound. However, the experiments could be made more thorough (please see the comments above). My main concern is the amount of computation which seems to be quite high. Even though communication is the bottleneck in FL, the amount of computation at each local client should also not be too high. An empirical evaluation of this aspect would be highly encouraged to demonstrate the practicality of the proposed methods on FL devices.

---

### Decision · Program_Chairs · 2023-01-20

**Decision:**

Accept: poster

**Justification For Why Not Higher Score:**

Some claims are misleading or not fully supported. Inadequate experimental evaluations.

**Justification For Why Not Lower Score:**

New state-of-the-art theoretical contributions.

**Metareview: Summary, Strengths And Weaknesses:**

Paper studies Federated Learning (FL) problem under an second order similarity (SOS) condition, which bounds the average deviation of the rate of change of gradient at a client from the global gradient's rate of change. Authors propose two new algorithms SVRP and Catalyzed SVRP. SVRP is a variance reduced version of an approximate stochastic proximal point method (SPPM). And Catalyzed SVRP is Catalyst-based accelerated version of SVRP. It is proved that these algorithms achieve better communication complexity than previously known methods under partial client participation and the SOS condition.

Non-approximate version of SPPM algorithm is known previously as APROX (Asi & Duchi, 2019). Authors claim
>  Theorem 1 for SPPM essentially gives the same rate as (Asi & Duchi, 2019, Proposition 5.3) but using constant rather than decreasing stepsizes and allowing approximate evaluations of the proximal operator.

However, this statement is misleading. APROX has diminishing stepsize $O(1/k^{\beta})$ [for all $0< \beta < 1$], but it is trivial to extend to $O(\epsilon)$ stepsize since the error decreases like $O(1/k)$. It is also non-standard to claim $O(\epsilon)$ stepsize is constant since it depends on the target sub-optimality $\epsilon$. Further, the approximation level required for the iterations is $b=O(\epsilon^3)$ (very small, practically restrictive) and this same small approximation error may be easily exploited in the known APROX analysis. For clarity, It is better to write the condition on $b$ in Theorem 1 explicitly in term of $\epsilon$ instead of $\eta$. In discussion with reviewer, authors claim that their new analysis for SPPM allows for use of variance reduction to exploit the SOS condition. To showcase the importance of the new analysis, authors are encouraged to show in detail why this cannot be done using prior analysis techniques of APROX.

Authors also showcase their empirical communication complexity on synthetic and small ridge regression problems and compared with other algorithms with known theoretical guarantees. However, many of the algorithms used in practice were not compared against stating that they do not have theoretical guarantees. But reviewers opined that these are important baselines regardless of their theoretical guarantees. Further, it was also suggested that more extensive experiments on complex datasets (logistic regression on image datasets) with biased partitions and neural network would further solidify the practical performance and verify whether the assumptions hold in practice. After all FL is field driven by practical considerations.

**Note From Pc:**

if the above contains the word "oral" or "spotlight" please see: "oral" presentation means -> notable-top-5% and "spotlight" means -> notable-top-25%. As stated in our emails, we are disassociating presentation type from AC recommendations